# A transient enhancement of Mercury's exosphere at extremely high altitudes inferred from pickup ions

Jamie M. Jasinski [1✉], Leonardo H. Regoli [2,3], Timothy A. Cassidy [4], Ryan M. Dewey [2], Jim M. Raines[2], James A. Slavin [2], Andrew J. Coates [5,6], Daniel J. Gershman[7], Tom A. Nordheim[1] & Neil Murphy[1]

Mercury has a global dayside exosphere, with measured densities of $10^{-2}\,cm^{-3}$ at ~1500 km. Here we report on the inferred enhancement of neutral densities ($<10^2\,cm^{-3}$) at high altitudes (~5300 km) by the MESSENGER spacecraft. Such high-altitude densities cannot be accounted for by the typical exosphere. This event was observed by the Fast-Imaging Plasma Spectrometer (FIPS), which detected heavy ions of planetary origin that were recently ionized, and "picked up" by the solar wind. We estimate that the neutral density required to produce the observed pickup ion fluxes is similar to typical exospheric densities found at ~700 km altitudes. We suggest that this event was most likely caused by a meteoroid impact. Understanding meteoroid impacts is critical to understanding the source processes of the exosphere at Mercury, and the use of plasma spectrometers will be crucial for future observations with the Bepi-Colombo mission.

[1] NASA Jet Propulsion Laboratory, California Institute of Technology, Pasadena, CA, USA. [2] Department of Climate and Space Sciences and Engineering, University of Michigan, Ann Arbor, MI, USA. [3] Applied Physics Laboratory, The John Hopkins University, Laurel, MD, USA. [4] Laboratory of Atmospheric and Space Sciences, University of Colorado Boulder, Boulder, CO, USA. [5] Mullard Space Science Laboratory, UCL, Dorking, UK. [6] Center for Planetary Sciences, UCL/Birkbeck, London, UK. [7] NASA Goddard Space Flight Center, Greenbelt, MD, USA. ✉email: jasinski@jpl.nasa.gov

Mercury has a tenuous exosphere, which is supplied by particles released from its surface. The composition of this exosphere is now known to contain H, He, Na, K, Ca, Mg, Al, Fe, and Mn[1]. The most continuously observed species by the MESSENGER spacecraft were Na, Mg, and Ca[2]. Sodium is the most abundant observed species in the exosphere and has been the most well studied from ground observations as well as by MESSENGER. This sodium exosphere has small dayside scale heights of up to ~100 km at perihelion, with subsolar densities of $10^3$ cm$^{-3}$ measured at altitudes of ~450 km[3]. Due to its proximity to the Sun, Mercury's sodium exosphere experiences radiation pressure which compresses the exosphere at the subsolar region and accelerates sodium atoms to escape velocities on the nightside to form a cometary-like tail[4-9]. Similarly, the magnetosphere is compressed on the dayside by the solar wind (SW), and the nightside magnetic field is stretched out to form a magnetotail, and so the exosphere mostly lies within the magnetosphere of Mercury.

Here we show an extreme event where the MESSENGER spacecraft observed newly ionized particles from a neutral cloud of exospheric particles at high altitudes. We analyze the pickup ion's velocity and infer the neutral densities from the observed ion fluxes. We conclude that the cause of the event is the impact of a meteoroid at Mercury, which vaporized Na and Si from the surface. These particles were subsequently photoionized in the solar wind and observed by the MESSENGER spacecraft.

## Results

**Event overview.** The MESSENGER spacecraft frequently spent time in the solar wind outside of Mercury's magnetosphere and the bow shock[10]. On December 21, 2013, MESSENGER measured unexpectedly large heavy-ion counts (of planetary origin) in the solar wind where only protons or alpha particles are usually measured. Figure 1 shows the location and trajectory of MESSENGER during these observations. The spacecraft was traveling northwards and was located in the SW outside of the bow shock (red line). The bow shock in Fig. 1a is in the noon–midnight meridian (i.e., at $Y = 0$). The spacecraft however, is positioned

dawnward of Mercury (i.e., $Y' \sim -2\ R_M$, where $R_M = 2440$ km) where the bow shock is northward of MESSENGER. Therefore, the appearance of the spacecraft inwards of the bow shock is a projection effect, and MESSENGER lies outside the bow shock in the solar wind as can more accurately be visualized in Fig. 1b, (where the bow shock boundary and the spacecraft are aligned in the same plane).

On the right (Fig. 1b), it can be seen that high densities (~$10^2$ cm$^{-3}$) of the sodium exosphere are located close to the planet, and with a small-scale height, the neutral density is extremely tenuous at altitudes >2000 km on the dayside and has background-level emission intensities outside the bow shock[3,11]. An ion that has just been ionized in the nominal exosphere where the high exospheric densities are distributed close to the planetary surface, would be located inside the magnetosphere. The ions that are observed in the solar wind cannot originate from inside the magnetopause. Any newly ionized planetary particle (e.g., sodium) will not escape the magnetosphere due to the stronger magnetic fields (close to the planet). Stronger planetary magnetic field magnitudes will keep a heavy planetary ions' gyroradius smaller (<0.1 $R_M$ for a 10 keV e$^{-1}$ Na$^+$ ion in a 300 nT magnetic field) than in the solar wind; prohibiting the heavy ion (sodium in this example) from gyrating out into the SW. Therefore, it is very surprising that we observe heavy ions of planetary origin in the solar wind.

A timeseries of the MESSENGER plasma and magnetic field observations can be seen in Fig. 2. Background-level proton fluxes were measured (Fig. 2a) in the solar wind. High counts of heavy ions (Fig. 2b) with a mass-to-charge ($m/e$) ratio of 21–30 amu/e were observed (starting at 00:26 UT) with energies centered at 10 keV e$^{-1}$ (close to the limit of the FIPS energy range of ~13 keV e$^{-1}$). Sodium (which lies within this $m/e$ range) is a major constituent of Mercury's exosphere and is easily photoionized to Na$^+$. In contrast, Na$^+$ in the SW are very rare, and, if present, would very likely be detected in a higher charge state. Therefore, these heavy ions are of planetary origin.

During the MESSENGER mission, FIPS' measurements of heavy ions were binned to increase the signal-to-noise ratio. Na$^+$

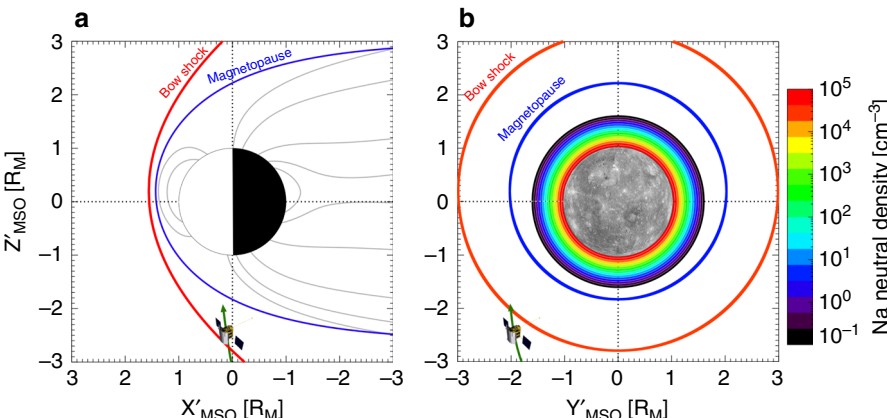

**Fig. 1 Spacecraft location.** Locations of the exosphere and magnetospheric boundaries, and the trajectory of MESSENGER during the heavy-ion observations. The coordinate system is in the aberrated Mercury solar orbital frame (MSO). **a** View from dusk ($X'$-$Z'$ plane) in the noon–midnight meridian, with the Sun to the left. The solar wind is to the left of the bow shock, and magnetospheric model field lines[51], help visualize Mercury's magnetosphere. The location of the spacecraft is seen to be inside the bow shock in (**a**), however the spacecraft has a large component of its position directed out of the plane, and is actually in the solar wind. **b** The view from the Sun ($Y'$-$Z'$ plane). Also shown is a simple sodium exospheric model with a scale height of 100 km which was the measured scale height at the subsolar point for a true anomaly angle (TAA) of ~180° (i.e., at aphelion) in Mercury's elliptical orbit[3,11]. This model exosphere is extremely simple and does not capture many of the details of the actual exosphere. The model is only intended to show how high neutral sodium densities (>$10^2$ cm$^{-3}$) are typically expected to be close to the dayside surface in comparison to MESSENGER's altitude of 5300 km. The bow shock and magnetopause were drawn using a model derived from MESSENGER observations, for the observed conditions for this event[28]. Source data are provided as a Source Data file.

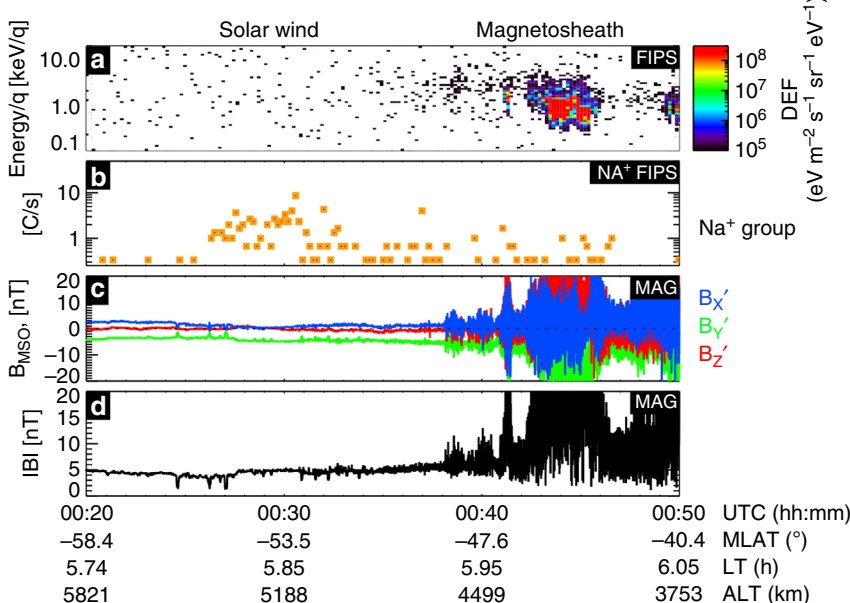

**Fig. 2 Spacecraft observations.** MESSENGER observations from FIPS and MAG on Dec 21, 2013. **a** FIPS proton differential energy flux (DEF); (**b**) FIPS measured sodium-group ion counts per second; (**c**) three components of the magnetic field in aberrated MSO coordinates; and (**d**) magnetic field magnitude. Source data are provided as a Source Data file.

measurements were binned with other species with similar mass-to-charge ratios in what is called the "sodium-group" ($m/e$ = 21–30 amu/e, including $Na^+$, $Mg^+$, $Al^+$ and $Si^+$). Therefore, it is not possible to directly distinguish between these different species, and any mention of sodium-group ions ($Na^+$-group), heavy ions, or pickup ions in this paper therefore refer to the above-mentioned group of species.

The event discussed in this paper had the highest $Na^+$-group count rate (1.2 counts $s^{-1}$) in the solar wind of the entire MESSENGER mission (for a 10 min bin), with the mean solar wind $Na^+$-group count rate being <0.01 counts $s^{-1}$ (see Methods, Sodium-group Ions in the solar wind for more details). Figure 2c, d show the magnetic field was unvarying during the observations, with a magnitude of ~5 nT and mainly orientated in the $-Y'$ direction (dawnward) at ~5 nT during the high $Na^+$-group count observation. Thirteen minutes after the peak heavy ion count rate is measured, MESSENGER crossed the bow shock and entered the magnetosheath.

**Pickup ion analysis.** Figure 3a shows the angular distribution of the pickup ions measured by FIPS. The pickup ions were observed to be flowing in the southward ($-Z$) direction—which is the direction of the solar wind motional electric field (**E**). Figure 3b shows where in velocity-space pickup ions are theoretically expected to be observed during the various stages of the pickup process. Upon photoionization, the ions are injected ('Inj') into the plasma, and are initially accelerated along the motional electric field which is described by $E = -V_{SW} \times B$. The ions then gyrate around the magnetic field (**B**). This distribution is unstable to waves that will eventually pitch angle scatter the ions into a bispherical shell distribution[12]. Figure 3c, d show the FIPS observations in the solar wind magnetic frame (SWB) in 2D projections of the theoretical velocity distribution shown in Fig. 3b, and use heliospheric RTN coordinates (see Methods, Coordinate Systems for more details), which are closely aligned to the SWB frame (during this event). $R'$ is in the solar wind velocity direction and points toward the planet (and is in the $-X'_{MSO}$ direction). $T'$ is the cross product of the Sun's spin angular

velocity vector and the $R$ vector (i.e., $T'$ is directed along $-Y'_{MSO}$), and N completes the right hand set (N points along $Z_{MSO}$).

In Fig. 3c the magnetic field is approximately in the $T'$ direction (dawnward) and only extends out of this plane (the ecliptic) by ~1°. In Fig. 3c, d, the shell distribution at the $\mathbf{w} = 1$ shell (where $\mathbf{w} = \mathbf{V_{ion}}/V_{SW}$) is shown by the dotted circle, and the expected gyrotropic ring distribution can be seen on the dashed line. The gyrotropic ring distribution (dashed line) bulk velocity is located at $(0, 0, \mathbf{V_{SW}} \cdot \mathbf{B}/B)$[12]. The direction of the electric field is calculated using $\mathbf{E} = -\mathbf{V_{SW}} \times \mathbf{B}$. The star shows the expected location of the injection site of newly ionized particles injected into the plasma, and is calculated using $-\mathbf{V_{SW}} \cdot (\mathbf{E} \times \mathbf{B})/|\mathbf{E} \times \mathbf{B}|$[13,14].

It is clear from the gray shading, that FIPS did not have a complete view of the theoretical ring distribution. However, from the region of velocity space that FIPS could sample, it is also clear that the measured pickup ions were highly localized in velocity space and not spread out along the expected ring distribution. Upon injection, the particles will first move along **E** and so an initial non-gyrotropic distribution will form and move anti-clockwise (as the ion gyrates around **B**) from the injection site (star) (Fig. 3d). The observation of a clumping of ions close to the injection site along **E**, provides evidence that FIPS measured non-gyrotropic pickup ions that were undergoing acceleration along the electric field, during the first gyration after injection.

A non-gyrotropic distribution would have its maximum phase space density (PSD) in the lower left quadrant going anti-clockwise (Fig. 3d) from the star. Any filling of other quadrants depends if there has been enough time and distance for the ions to make a complete gyration and become dispersed in a ring. Gyrotropic ring distributions are largely found at highly active comets where there is a large source of neutrals for ionization and a large enough interaction region (several orders of magnitude larger than the ion gyroradius). Non-gyrotropic rings have been observed at weak comets such as 26P/Grigg-Skjellerup where the ion production rate varies with distance due to the varying neutral density, and the density varies with gyration angle throughout the cycloid motion of the pickup ion[12]. Similarly, at such large altitudes (in the solar wind) where Mercury's global exospheric density is negligible, any enhancement will be

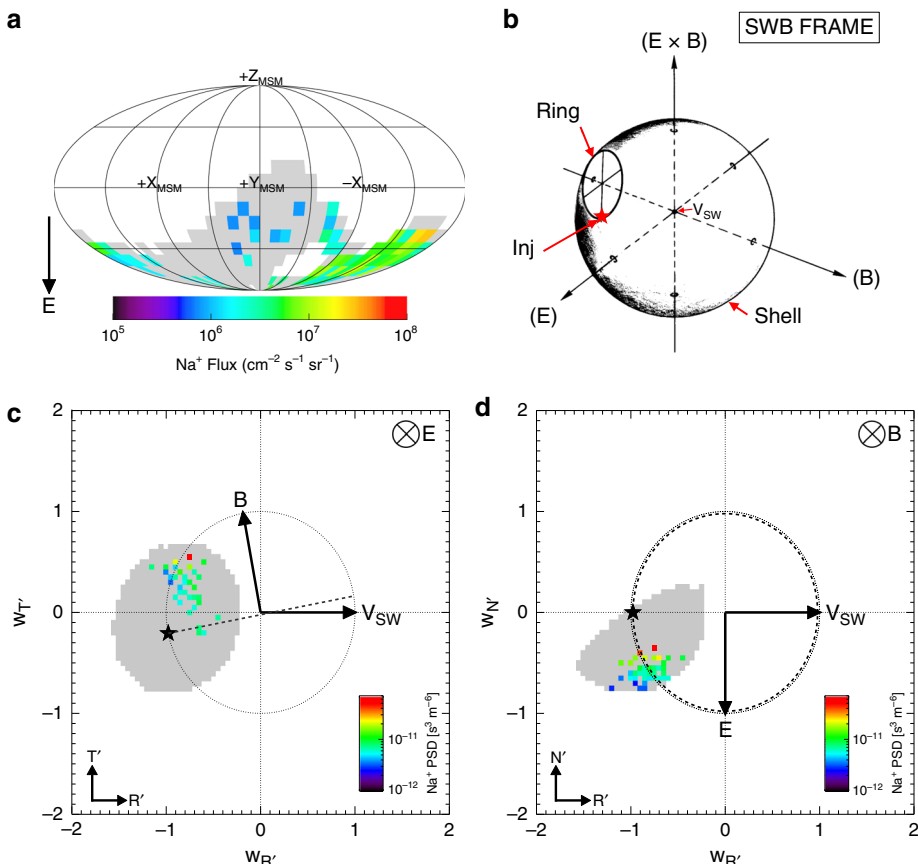

**Fig. 3 FIPs observations of pickup ion distributions.** Model and observed pickup ion distributions in the solar wind (outside the bow shock). **a** Shows the angular flux map of the observed sodium-group pickup ions in the spacecraft frame; (**b**) shows the 3D schematic of the expected shell and ring distributions in the solar wind magnetic ('SWB') frame[12]; (**c**, **d**) the measured pickup ion phase space density ('PSD') for ion measurements above the 1 count level in 2D plane projections of the SWB frame (i.e., **b**), in solar wind velocity units, where $\mathbf{w} = \mathbf{V_{ion}}/V_{SW}$. The data are shown in aberrated RTN coordinates ($R$ is directed in the solar wind flow direction; $T$ is Mercury's orbital direction and $N$ is positive northward with respect to Mercury) which are described in detail in the Methods, Coordinate Systems section. The dotted and dashed lines represent the shell and ring distributions respectively. The star shows the expected injection location of implanted ions in the plasma. The arrows show the directions of the magnetic (**b**) and electric fields (**e**) when they lie within that plane (and into the plane as a crossed circle). The gray shading (in **a** and **c–d**, respectively) show the region of velocity space sampling that the FIPS instrument covered but did not detect pickup ions. Source data are provided as a Source Data file.

intermittent and therefore we do not expect a large interaction region for the pickup process.

The energy of the pickup ions measured by FIPS was found to have a distribution of 9–13 keV per e, which is at the upper limit of the instrument's energy range (0.046–13 keV e$^{-1}$). The ions will initially have very low energies; however, it is the pickup process that transfers energy from the solar wind into the newborn ions and accelerates them to the energies we observe here. This acceleration is initially completed by the solar wind motional electric field, and the maximum energy is dependent on the angle between the magnetic field and $\mathbf{V_{sw}}$. At comets for a $V_{SW} = 440$ km s$^{-1}$, $H_2O^+$ can be accelerated up to energies of 70 keV e$^{-1}$ for an angle of 90° [15]. During our event, the motional electric field is estimated to be 0.002 V m$^{-1}$, and this electric field is high enough to accelerate a planetary pickup ion to the observed energies within ~30 s (from rest up to 10 keV e$^{-1}$ for Na$^+$, Mg$^+$, Al$^+$ and Si$^+$). This is much smaller than the estimated gyroperiod of 5–6 min (for this group of species). This would mean that the pickup ion would be expected to be observed in the lower left quadrant of Fig. 3d (as not enough time has passed for the ion to make a full gyration), which is where it is detected. The pickup process would eventually accelerate the ions to much higher energies than observed. We are however, detecting these

ions midway through this acceleration process, and therefore we do not expect that we are missing much of the ion distribution (due to FIPS' restricted field of view) considering the ions are observed as a localized non-gyrotropic beam.

**Neutral density estimation.** To understand more about the origins of the pickup ions and the size of the interaction region during this event we have back-traced the ions in time[16,17], and then estimated the neutral densities required to produce the pickup ion flux measured by FIPS. The interaction region, which we are trying to determine, is the approximate size of the neutral cloud (or plume) that the ions are coming from. The size of the region will affect the ion flux that is observed (discussed below). Figure 4a, b shows the results of the particle tracing effort. Each colored line shows the back tracing of a pickup ion detected by FIPS. This was completed under the solar wind conditions measured by the magnetometer (MAG) (magnetic field largely in the $-Y$, or $T$ direction). This does not include the increased magnetic field strength and variability in the magnetosheath and also does not include the magnetospheric environment of higher magnetic field strengths. The aim of the tracing effort is to estimate the size of the interaction region where the ionization took place. We do not expect this to be far from the spacecraft; because

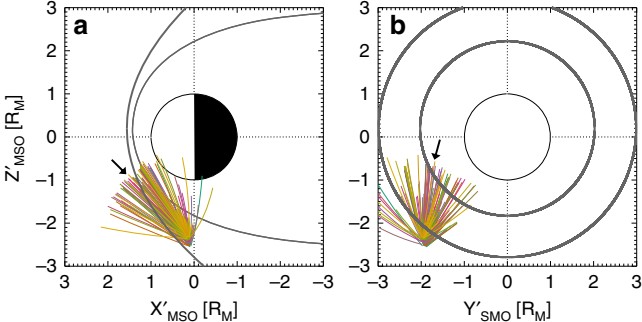

**Fig. 4 Particle tracing results.** Maps of the particle back-tracing of ions measured by FIPS (colored lines), with the inner and outer gray lines showing the location of the magnetopause and bow shock respectively. In all cases, the particles are traveling toward the spacecraft and the arrow is an example of that direction. **a** Shows the $X'$-$Z'$ plane, and (**b**) shows the $Y'$-$Z'$ plane. Source data are provided as a Source Data file.

the ions were observed shortly after ionization (~30 s) and the non-gyrotropic nature of the pickup ions (which had a gyro-period of 5–6 min) means this distance should be significantly smaller than the pickup ion gyroradius of 5.6–6.2 $R_M$ (for a pickup ion in a 5 nT magnetic field that is either $Na^+$, $Mg^+$, $Al^+$, or $Si^+$). Therefore, we limit the tracing to 2 $R_M$ and outside the bow shock.

Trajectories that appear inside the bow shock in Fig. 4 are due to the projection of the bow shock in a 2D plane and the tracing having a component in or out of the page. We can see that from the particle tracing the ions mostly originate upstream from the spacecraft within a region within 0.5–1 $R_M$ north of the observations. The ions would not originate from the exosphere close to the planet, because the intense magnetic fields of the magnetosphere would result in extremely small gyroradii of the ions. At the surface, magnetic fields of ~300 nT result in a pickup ion gyroradius of ~0.1 $R_M$ (assuming the ion's velocity is perpendicular to B and the ion is either $Na^+$, $Mg^+$, $Al^+$, or $Si^+$). At the limits of the magnetosphere where the field strength is ~80 nT, the gyroradius is 0.4 $R_M$. Both values are much too small for the ions to be measured in the solar wind at MESSENGER's altitude, if they originated within the magnetosphere.

The average measured pickup ion integrated flux measured by FIPS is ~$10^{10}$ m$^{-2}$ s$^{-1}$. To observe these fluxes we estimate that the required neutral density is expected to be ~$10^2$ cm$^{-3}$ if we assume the ions detected are $Na^+$, $10^1$–$10^2$ cm$^{-3}$ if we assume $Si^+$, and ~$10^0$ cm$^{-3}$ if we assume $Al^+$ (see Methods, Estimating Neutral Densities for more details). For all species these are density values that would not be expected to be observed at such large altitudes. For sodium, these values are similar to sodium exospheric densities found at ~700 km altitude (for the high-density subsolar point 3[18]), and therefore at ~5300 km altitude the global exosphere cannot account for such high sodium densities (if we assume a composition solely of $Na^+$ in our measurements). To explain this observation, requires a nontypical interpretation.

## Discussion
There are four main processes that account for the neutral exosphere (which is of planetary origin): thermal desorption, photon-stimulated desorption, ion sputtering, and impact vaporization. Thermal desorption is when atoms are released from a surface owing to heating and is a function of the vibrational frequency of the atom and its binding energy to the surface. Thermal desorption produces atoms with energies less than the required energy of ~1.5 eV to reach high altitudes of 5300 km, and therefore is not

considered further. Photon-stimulated desorption occurs when an electron transfer is induced by the bombardment of photons, which then can desorb the atom from the surface. This process however, produces neutrals with a distribution of energies centered at low escape velocities. A very small fraction of the neutrals from photon-stimulated desorption will be at high altitudes[19] (also not considered further). Thermally desorbed or photo-desorbed atoms are much less likely to reach such high altitudes as compared to impact vaporization or sputtering. The low-energy portion of the Na exosphere, for example, is typically confined to within about 1000 km altitude.

Ion sputtering occurs when an ion impacts the surface and transfers energy that results in another particle being released. Sputtering produces particles with the highest escape rates[19]. At Mercury, sputtering is due to solar wind-magnetosphere coupling at the magnetopause, where magnetic reconnection energizes solar wind plasma and injects it into the magnetospheric cusps to precipitate onto Mercury's surface[10,20–23]. However, for 3 h preceding this event the IMF was quiet. The magnetic field magnitude was low at ~8 nT (average magnitude is ~15 nT[24]) and was mostly orientated sunward with observed mean values of: $B_X = 5.5$, $B_Y = -1.4$, $B_Z = 0.9$ nT. In contrast large IMF strengths orientated in a southward ($-B_Z$) direction are more conducive to intense reconnection at the dayside that will produce higher field-aligned electric fields, which will inject particles (with field-aligned pitch angles) that will precipitate onto the surface[25–27]; something we do not observe here. Furthermore, there are no reported coronal mass ejections or solar energetic particle events observed at this time by MESSENGER[28,29]. Therefore, sputtering is not considered to be the cause of the observation here.

Finally, the process of impact vaporization seems the most likely for the event observed here. Impact vaporization occurs when an object, such as a meteoroid strikes the surface of the planet and liberates particles from the surface (which is commonly observed at the Moon[30]). As a source for the sodium exosphere, impact vaporization is highly debated, and is estimated to account from as little as 1% of the total contribution to the exosphere[31] and up to 20% of the photon-stimulated desorption contribution[32]. In contrast, impact vaporization has been postulated as the dominant source of both the Ca and Mg exospheres at Mercury[33,34]. In any case, an impact will vaporize the sodium-group species with high enough energies that a fraction of the atoms will reach the altitude of the FIPS observation. One possible source for an impact could be from a crossing of comet Encke's stream (TAA ~ 130–160°[35]), which Mercury completed crossing a week earlier, however an asteroid source is more likely to produce a single large impact. Objects from the main asteroid belt can provide meteoroids owing to the 3:1 and v6 resonances, which deflect them into the inner solar system. The occurrence of an object up to 1.5 m in radius is expected to be 2 per year[36]. A previous model of the sodium exospheric density response to an impact by a 1 m meteoroid[36], found that at 1500 km altitude the sodium neutral density would be $10^4$ cm$^{-3}$ [adjusting for their incorrect surface sodium abundance by an order of magnitude[37]]. Extrapolating their observations to 5300 km, and assuming that the surface composition in the southern hemisphere is similar (to an order of magnitude) to the north, suggests neutral sodium densities in the range of ~$10^2$ cm$^{-3}$ would be expected. This is within our estimated density range (from FIPS observations). Finally, we estimate that the composition of the pickup ions is most likely a combination of Na and Si (please see the Methods, Composition Estimation section for more details). Al has the highest ionization frequency of $6 \times 10^{-3}$ s$^{-1}$ (i.e., lowest photo-ionization lifetime of 170 s) and so the neutrals would mostly have been ionized before reaching high altitudes. Mg has a very low ionization frequency ($5.4 \times 10^{-7}$ s$^{-1}$, or a photoionization

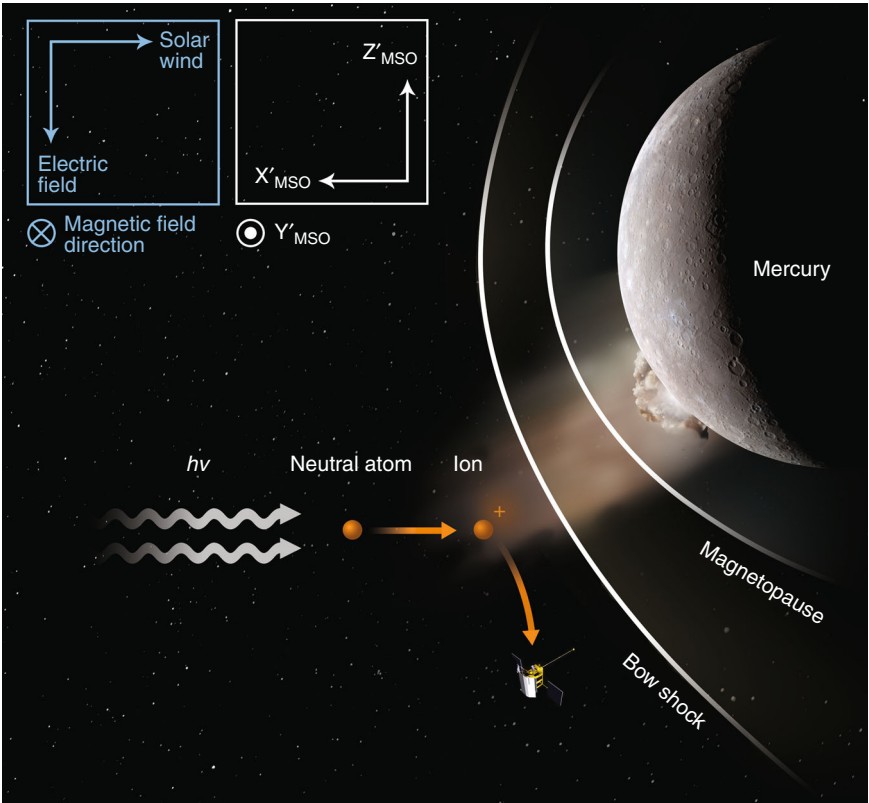

**Fig. 5 Illustration of the event.** A schematic (not to scale) of the pickup ion process occurring during our observed transient exosphere event. *hv* shows photons, which ionize the planetary atoms originating from a plume caused by a meteoroid impact event. These ions are then picked up by the solar wind and gyrate around the interplanetary magnetic field.

lifetime of $10^6$ s) and therefore not enough ions would be produced to provide the observed ion flux. Therefore, Si and Na are the most likely candidates to have produced the observed pickup ions.

In conclusion, the most likely cause of our sodium-group pickup ion event is meteoroid impact vaporization which caused a plume of Na and Si to extend far from the planet's surface. A schematic of this process occurring at Mercury can be seen in Fig. 5. With the arrival of Bepi-Colombo, there will be further possibilities to detect such enhancements by the plasma instrumentation, which will be crucial in understanding this phenomenon more.

## Methods

**Coordinate systems**. The magnetic field measurements and the data from MAG are presented in Mercury solar orbital (MSO) coordinates. $X$ is in the planet-Sun direction, $-Y$ points toward planetary orbital velocity vector direction, and $Z$ completes the right-hand set and points northward.

The coordinates and the data are aberrated to account for the high orbital velocity of the planet (39–59 km s$^{-1}$). This high orbital velocity results in the effective solar wind arrival vector being offset by ~7° from the $X_{MSO}$ direction. The orbital velocity varies over the Mercury's year due to its highly eccentric orbit. Here we use the aberration determined by Boardsen et al. (2010)[38], which calculated the aberration angle for each day, and use the appropriate aberration for this event. A vector that has been aberrated can be identified by an apostrophe (e.g., $X'_{MSO}$)

We have presented the pickup ion data in aberrated RTN coordinates where $R'$ is in the solar wind velocity direction and points toward the planet. $T$ is the cross product of the Sun's spin angular velocity vector and the $R$ vector, and $N$ completes the right hand set. The RTN coordinate system corresponds to the XYZ (MSO) system whereby $R \sim -X$, $T \sim -Y$, and $N \sim Z$. In Fig. 3b) we have shown the projection of the magnetic field (**B**) in the $R$-$T$ plane.

**Instrumentation**. Data from two instruments onboard MESSENGER (MErcury Surface, Space ENvironment, GEochemistry, and Ranging spacecraft) were used to analyze this event: the Fast Imaging Plasma Spectrometer (FIPS)[39] and the MAG[40].

FIPS was a time-of-flight mass spectrometer. The instrument measured ions with a range in energy-per-charge of 46 eV–13 keV e$^{-1}$, a range in mass-per-charge of 1–60 amu/e, and a time resolution of ~8 s. The angular resolution was ~15°. The effective field of view of FIPS was ~1.15 $\pi$ steradian as 0.25 $\pi$ steradian was blocked by the spacecraft sunshade. Because FIPS does not have a 4$\pi$ steradian field of view, unless the boresight of the instrument is pointing close to the solar wind ram direction, solar wind plasma will not be observed (as was occurring during this event). The signal-to-noise of heavy ions is improved by grouping heavy planetary ions into two groups: the Na$^+$ group ($m/e = 21$–30 amu/e, including Na$^+$, Mg$^+$, Al$^+$, and Si$^+$), and the O$^+$ group ($m/e = 16$–20 amu/e, including O$^+$ and water-group ions). This process has been described in significant detail[41].

MAG was a fluxgate MAG. It was mounted on a 3.6 m long boom. It had a resolution of 0.047 nT and a maximum time resolution of 20 vectors per second.

**Sodium-group ions in the solar wind**. When averaging over a 10 min window, for all the observations in the solar wind, this event has the highest sodium ion count rate, at 1.2 counts s$^{-1}$. High sodium count rates are not common in the solar wind, with the mean being 0.007 c s$^{-1}$. There are a total of 14 events in the solar wind (during 4 years of observations by MESSENGER at Mercury), with a count rate of higher than 0.4 c s$^{-1}$.

**Ion cyclotron waves (ICWs)**. Ion cyclotron waves are usually observed with pickup ions, which are observable in magnetic field fluctuations. We performed fourier analysis of the magnetic field data but did not find any evidence for ICWs. This is most likely due to the interaction region being very small in comparison to the sodium ion gyroradius. Any ICW will most likely be observed far downstream of our observations.

**Solar wind and spacecraft frame**. The data were transformed into the solar wind plasma frame (SWB), shown in Fig. 3. This was accomplished by subtracting the solar wind bulk velocity vector from the data. Due to an incomplete field of view, FIPS did not measure SW protons, and so $\mathbf{V}_{SW}$ cannot be estimated. Therefore, we use a nominal solar wind speed of 440 km s$^{-1}$ [10] taken at Mercury, which corresponds to a 1 keV proton, which is normally measured by FIPS (similar to previous studies when the solar wind is unknown[10,27]), which results in a vector direction in the aberrated RTN frame of (440, 0, 0) km s$^{-1}$.

**Table 1 Composition Analysis. Plume parameters for different species that could be detected in the sodium-group ions by FIPS. "Plume Density" is from Mangano et al., (2007)[36] and "Surface Composition correction" is from McCoy et al., (2019)[46].**

| Element | Photoionization lifetime (s) | Photoionization frequency ($s^{-1}$) | Plume density at 1500 km ($cm^{-3}$) | Surface Composition correction | Ion production rate at 1500 km ($cm^{-3}\,s^{-1}$) | Comments for 5300 km altitude |
|---|---|---|---|---|---|---|
| Na | $3 \times 10^4$ | $4 \times 10^{-5}$ | $4 \times 10^2$ | 10 | 0.1 | |
| Mg | $2 \times 10^6$ | $5 \times 10^{-7}$ | $4 \times 10^3$ | 0.5 | $10^{-3}$ | |
| Al | 170 | $6 \times 10^{-3}$ | $3 \times 10^2$ | 2–4 | 3.5–7 | Photoionized before 5300 km |
| Si | $5 \times 10^3$ | $2 \times 10^{-4}$ | $1.5 \times 10^4$ | 1 | 3 | Plume density will fall off much faster with altitude. Si more likely to be bound as SiO. |

**Not the foreshock**. The measured ions are not likely to be caused from quasi-parallel bow shock processes (where the IMF is parallel to the normal of the bow shock), such as the foreshock (observed at the terrestrial bow shock[42]). Although the event takes place on the dawn side of the bow shock, and the IMF is orientated dawnward, due MESSENGER's location southward of the planet, the bow shock conditions at the boundary crossing are quasi-perpendicular (therefore opposite to the conditions which accelerate particles at the foreshock). Furthermore, a measurement of the foreshock population by FIPS would also require high proton fluxes alongside sodium ion observations, something we do not observe.

**Particle tracing of test particles**. Due to the non-gyrotropic nature of the observed distribution, the trajectories of the sodium-group ions can be studied. Because we expect the observed ions to originate from outside the magnetosphere, a simple uniform background electromagnetic field is used, based on the instantaneous measurements provided by MESSENGER.

The particle tracing, which has been applied to study the interaction of Titan with the Saturnian magnetosphere[16,17] makes use of a 4th-order Runge Kutta algorithm to integrate the Lorentz force. For each location of MESSENGER during the period being analyzed, a particle is launched using a negative time step (backtracing) and a vector with inverted components with which particles were detected as initial velocity from FIPS. This backtracing approach allows us to identify the location (local to MESSENGER) from where particles might be coming from, using:

$$\mathbf{F} = q(\mathbf{E} + \mathbf{v} \times \mathbf{B}). \tag{1}$$

Although no small-scale variations in the background field are considered (because measurements are available in-situ only), the large gyroradius of the particles compared with the size of the interaction region means that small variations would play a negligible role in deviating particles from the calculated trajectory. In addition to this, the orientation of the interplanetary magnetic field during the short time studied here is not expected to change significantly[24].

Once the particles are initialized, their trajectory is followed until they reach the nominal location of the magnetopause. Due to the non-gyrotropic nature of the ions traveling along the E-field, we do not expect the interaction region to be very large. We also do not expect the sodium-group enhancement to be on a global scale, so we do not expect the interaction region to be >2 $R_M$ (so expect it to be southward of the equator), which is the upper estimate of further analysis.

**Estimating neutral densities**. At aphelion of Mercury's orbit, the sodium photoionization lifetime is expected to be 8.5–10.5 h, taken from the theoretical published lifetimes[2,43,44]. This corresponds to a sodium ionization frequency ($\lambda$) of $\sim 3.6 \times 10^{-5}\,s^{-1}$. The ionization lifetimes and frequencies for all the possible species (i.e., $Na^+$, $Mg^+$, $Al^+$ or $Si^+$) in the sodium-group are summarized in Table 1. The ionization frequency, $\lambda$, is a useful parameter as it can be used to calculate ion production rate simply by multiplying it with the neutral density ($n$), for altitudes well above the exospheric peak[45].

Using $flux = \lambda n\, v/a$ (where $\lambda n$ is the ion production rate, $v$ is the volume of the interaction region to estimate the total number of ions produced, and $a$ is the area through which the ions are traveling through along the electric field) we can estimate $n$. Therefore, the size of the interaction region (i.e., the size of the neutral cloud where the particles are being ionized) is important, as it will affect the ion flux that we observe.

Modeling the neutral cloud as a simple sphere, box, or cylinder with a radius or length of 0.5 $R_M$, gives us the number of particles inside the cloud by multiplying by the volume. Multiplying by the ionization frequency then gives us the number of ions being produced per second. These ions then begin to flow in the direction of the electric field, out of the neutral cloud (anisotropically) through an area dependant on the size of the cloud, to produce an ion flux (i.e., $flux = \lambda n\, v/a$). Therefore, the ion flux is dependent on both the size of the cloud and the density. By constraining the size of the cloud (from the observations and the particle tracing effort) we can estimate the density of the neutral cloud. We estimate that the neutral densities required to produce the ion fluxes (that we observed to be $1.5 \times 10^{10}\,m^{-2}\,s^{-1}$) are $4 \times 10^2\,cm^{-3}$ if we assume $Na^+$; $2.3 \times 10^4\,cm^{-3}$ if we assume

$Mg^+$; 2.1 $cm^{-3}$ if we assume $Al^+$; and 56 $cm^{-3}$ if we assume $Si^+$. With this simple function, for a fixed flux, $n$ is dependent on the length scale $x$ as $v/a \sim x$. With a larger interaction region (e.g., 1 or 2 $R_M$), the required neutral density is lower but still $\sim 10^2\,cm^{-3}$ for Na; $\sim 10^4\,cm^{-3}$ for Mg; $\sim 1\,cm^{-3}$ for Al and $\sim 10^1\,cm^{-3}$ for Si (for both 1 and 2 $R_M$).

**Composition estimation**. Even though the ion measurements are binned in mass-per-charge ($m/e = 21$–30 amu/e, including $Na^+$, $Mg^+$, $Al^+$ and Si), we can make an educated guess as to what the most likely composition is of the observed ion signal.

Using the TOPBase estimations for the ionization frequency for an active sun from the tables shown in Huebner & Mukherjee, (2015)[44] and scaling these values for Mercury's heliocentric distance of 0.467 AU at a True Anomaly Angle of 177° the photoionization lifetimes and frequencies are shown in Table 1.

Also shown in Table 1 are the plume density estimates from Mangano et al., (2007)[36] at 1500 km altitude. We also show a surface composition correction factor that should be applied to the Mangano et al., (2007)[36] values, using latest surface composition from McCoy et al., (2019)[46]. The expected ion production rate at 1500 km altitude is shown, which is calculated by multiplying the plume density by the correction factor and dividing by the photoionization lifetime. From this calculation, Mg is expected to be a negligible component of our pickup ions composition.

Al has a lifetime against photoionization of <3 min, and so only a small fraction of Al would be expected to reach high altitudes to be observed by FIPS in the solar wind at high altitudes of 5300 km. Therefore, Al would not be expected to be measured by FIPS.

Finally, the 1500 km altitude ion production rate Si/Na ratio is ~30. However, the Si plume density is expected to decay more strongly than Na (due to its higher mass). Therefore, we expect this ratio to drop at high altitudes.

We also note that there is evidence for the possibility that meteoroid impacts release molecules that subsequently photodissociate into the neutral atoms that are observed at higher altitudes (e.g., for the Ca exosphere[47,48]). This may be an important process in regards to both Al and Si, with the photodissociation of AlO and SiO[49], and therefore we briefly discuss it here. Berezhnoy & Klumov (2008)[50] investigated the photodissociation lifetimes of these molecules and found them to be 2000 and 600 s (AlO and SiO, respectively). Due to the increased mass of the particle, these particles will have a lower velocity and will be photodissociated before reaching higher altitudes, in comparison to the atoms. After photodissociation, Al still has a very short photoionization lifetime, and so it will be ionized quickly after photodissociation. Therefore, we do not think this is a viable method for Al to reach high altitudes to be observed by FIPS—it will not reach extremely high altitudes in 2000 s as a molecule, and then it will be quickly ionized as an atom and trapped in the magnetosphere. This however, may be a more viable method for Si, because of the lower photodissociation lifetime (the molecule will photodissociate shortly after ejection from the surface), and with the longer photoionization lifetime it may be able to reach higher altitudes as an atom. However, investigating this process in depth is beyond the scope of this paper, and we do not consider it further.

In conclusion, we estimate that the FIPS ion composition is most likely a combination of both Na and Si.

**Reporting summary**. Further information on research design is available in the Nature Research Reporting Summary linked to this article.

## Data availability

All the data used in this study can be found at NASA's Planetary Data System. MAG data can be found at https://pds-ppi.igpp.ucla.edu/search/view/?f=yes&id=pds://PPI/MESS-E_V_H_SW-MAG-3-CDR-CALIBRATED-V1.0.

FIPS data can be found at https://pds-ppi.igpp.ucla.edu/search/view/?f=yes&id=pds://PPI/MESS-E_V_H_SW-EPPS-3-FIPS-CDR-V1.0/DATA.

Figures were created using IDL, Powerpoint and Photoshop. Source data are provided with this paper.

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

## Acknowledgements

The authors would like to thank Jacek Zmarz for producing the illustration shown in Fig. 5. J.M.J.'s contribution to this research was supported by an appointment to the NASA Postdoctoral Program (NPP) at the Jet Propulsion Laboratory administered by Universities Space Research Association through a contract with the National Aeronautics and Space Administration (NASA). L.H.R.'s contribution was funded by a NASA Living with a Star Grant (NNX16AL12G). A.J.C. acknowledges STFC support via the solar system consolidated grant to UCL-MSSL. N.M. acknowledges support from the Jet Propulsion Laboratory, California Institute of Technology, under a contract with NASA, as well as the NASA Discovery Data Analysis Program Grant (80NM0018F0612). The image of the MESSENGER spacecraft used in Figs. 1a, b and 5 are credited to NASA/ Johns Hopkins University Applied Physics Laboratory/Carnegie Institution of Washington.

## Author contributions

All authors were involved in the writing of this paper. J.M.J. led the work, identified the event, and conducted most of the analysis of this dataset. L.H.R. led the particle tracing effort and contributed to the determination of the inferred neutral densities. T.A.C. led the exospheric interpretation as well as the composition analysis. R.D., J.M.R., J.A.S. and D.G. provided knowledge of the FIPS instrument as well contributed to the analysis of the ion measurements. A.J.C., T.A.N. and N.M. provided expertize in the pickup process and contributed to the analysis of the ion data.

## Competing interests

The authors declare no competing interests.
