## [Peer Review File · Nature Communications]

Reviewers' comments:

Reviewer #1 (Remarks to the Author):

Summary

The observation of enhanced Na⁺ ions (really an enhancement of the “Na-group” ions – see below) at high altitude near the bow shock is indeed an interesting measurement. This can reasonably be viewed as coming from an impact event. However, the authors are too focused on Na in their explanation and they fail to consider both alternative sources (Mg) as well as to speculate on the timely nature of a potential impactor from comet Encke. As such, the paper needs some reworking in order to be published.

Major comments

The grouping of FIPS ions (lines 249-254) points to a particularly interesting possibility not considered by the authors – that this observation pertains to Mg and not to Na. Although it is true that Na is the dominant neutral, Mg has a lifetime against photoionization that is an order of magnitude longer than Na and thus can reach higher altitudes before experiencing photoionization. The temperatures necessary for the Mg distribution observed by MESSENGER (Merkel et al. 2017) already suggest that micrometeoroid impact vaporization is the likely primary source of the Mg exosphere. Column densities of Mg (Merkel et al. 2017) compared to Na (Cassidy et al. 2015) show Na is about 10 times that of Mg, so if the Na source rate from impacts is < 20% of the total Na, Mg is an equally plausible atom. Thus, an impact may have generated a cloud of Mg rather than Na (or more likely a cloud of both). It is premature of the authors to assume this is just Na⁺ rather than Mg⁺ or some combination of the two given the limitations of the FIPS data. This discrepancy cannot be ignored, so the paper is really about the observation of a potential impact event and not strictly about the Na exosphere. I suggest a title change to that effect, and I suggest that all references to Na throughout be changed to Na-group data or Na-Mg data or something of the sort. This reworking does not diminish the observational data, nor the fact that this is an interesting observation, but simply posits an explanation that is more likely and truthful to the situation. [Note that in the remaining comments, I refer to "Na" but I am speaking of the "Na-group ions". The language is just easier to type.]

The paper also neglects to consider the timing of this event. Mercury had a particularly close passage to comet Encke in 2013 only 3-4 weeks before this observation. Based on simulations from Christou et al. (2015), the Encke stream not only crosses the orbit of Mercury near TAA ~ 25 degrees (where

an increase in the Ca exosphere is observed), but it also crosses near TAA 130-160 degrees. This is not too far from the TAA ~ 177 degrees at which the observation occurred. The event also appears to happen on the dawn side of the planet (line 294), which is the ram direction of Mercury as it plows through the Encke stream. Because the Christou et al. work was primarily focused on smaller grains that are likely behind the Ca enhancement at Mercury, it is possible that the larger bodies needed for such an impact event as noted here would evolve under dynamic effects to slightly different TAAs. Either way, the timing may not be entirely coincidental and should be noted.

Specific comments

Abstract

The characterization of the exosphere as “closely bound” is a little misleading (line 18). The primary source of Na, photon-stimulated desorption (PSD), does indeed lead to the low-altitude Na exosphere having a small scale height. However, all the observed Na profiles extend to much higher altitudes (duly noted by the authors), with this higher-altitude component suspected of deriving from a higher energy source than PSD such as sputtering or micrometeoroid impact vaporization as only they have the energy to eject Na atoms with the velocities needed to reach these altitudes. The altitudes reached are a significant fraction of the planetary radius, so the Na exosphere is not quite as “closely bound” as the initial statement would indicate. This also completely neglects the fact that the Na tail (mentioned in the main text) is an extension of the Na exosphere that can extend many planetary radii antisunward. Reworking the opening statements is needed to be faithful to what is really the case. It does not detract from the point that neutral Na at 5300 km would be significantly less dense under normal circumstances.

The description of these as the first “measured enhancement of neutral sodium densities” (line 20) is incorrect. FIPS did NOT measure neutrals, so at best these are the first observations that INFER or SUGGEST an enhancement of neutral sodium densities.

The statement that “no observations attributable to micrometeoroids were detected” (lines 33-34) is not true. There are no published results connecting the MESSENGER *sodium* data to micrometeoroids (although the published densities at higher altitudes suggest that as a source), but there have been several papers detailing the connections of calcium to micrometeoroids. The blanket statement in the abstract should be more representative of the situation.

Main text

Line 43: The statement that the exosphere is “constrained within” the magnetosphere suggests that the magnetosphere is limiting the extent of the exosphere. I suggest that this be reworded such that the bulk of the Na exosphere simply lies within the magnetosphere.

Figure 1 caption: Under the description of panel (b) it says the “View from dusk”. If the Sun is to the right, then this is the view from *dawn* unless the planet is upside down (i.e., north is down). I cannot specifically say how the authors defined the MSO system to know definitively which way positive Z points, but I would suspect +Z is in the north direction and thus “up”. Note that I am only claiming this view is from dawn and not that any of the subsequent descriptions related to dusk in the text are incorrect.

Figure 1, panel (c): Either capitalize “neutral” or don’t capitalize “Density”.

Line 73: The spacecraft location in Figure 1b is inside the bow shock whereas it says here that the spacecraft crossed the bow shock and entered the the magnetosheath. I realize this is probably the difference between a model bow shock in Figure 1b and reality, but the disconnect is disconcerting and I recommend some rewording to make things clearer.

Figure 3 caption: For the uninitiated, it would be useful to define what wT' , wN' and wR' are here; at the very least point them to the “Methods” section where they are defined.

Line 110: I suggest “highly active comets” rather than “more active comets”; Mercury is not a comet.

Lines 150-151: Awkward sentence; rephrase.

Line 153: Hyphen needed in “photon stimulated”.

Line 155: “Due to” should be “owing to”.

Line 165: “Due to” should be “owing to”.

Lines 182-183: An example is given for the Moon, but there is ample published evidence that micrometeoroids at Mercury are liberating Ca, particularly during the passage of the comet Encke dust stream across Mercury's orbit (Burger et al. 2014, Killen and Hahn 2015, Christou et al 2015) where the source rate enhancement is quite large. It is also likely that the process is contributing to the Na exosphere at Mercury but the signal is smaller compared to the dominant PSD source rate. Nevertheless, there is no need to invoke the Moon here.

Lines 183-184: The numbers from Cremonese et al. (2005) and Burger et al. (2010) should be noted as pertaining specifically to a source of the Na exosphere. As just noted, the contribution of impacts to the Ca exosphere is both dominant and large. One cannot think of Mercury as having a single exosphere anymore.

Line 188: "micrometeorites" should be "micrometeoroids".

Line 188: "Due to" should be "owing to".

Line 192: "exospheric" should be "exosphere".

Line 196-197: It should be noted that if this Na comes from the southern hemisphere, the "measurements" of Na in Peplowski et al. (2014) are not true measurements but simply an assumption that the south is like the north owing to symmetry. However, that conclusion is based on the assumption that Na "migrates" to the poles; it is still possible that the hemispheres are just different compositionally. BepiColombo will hopefully shed light on this. Regardless, MESSENGER was too far from the southern hemisphere for the surface content of Na to be measured there. Thus, the order of magnitude difference cited relative to Mangano et al. (2007) may not be correct and that caveat should be noted.

Lines 197-199: Awkward sentence with badly placed commas; please rephrase.

Lines 200-206: The tail extension occurs at two points in Mercury's orbit, not just one (it's just that because of the motion of Mercury the tail is bigger/stronger on one part than the other). It is true that the tail at aphelion is small to nonexistent, so the point about trajectories being less altered is correct. That's all that needs to be said, but if the point about the tail is to be kept, it needs to be restated to be factually correct.

Line 253: The “[Killen et al., 2018]” reference should be “[Killen et al., 2019]”.

Line 304: “Since” should be “Because”.

Reviewer #2 (Remarks to the Author):

This paper reports original measurements performed by the Fast Imaging Plasma Spectrometer (FIPS) on board MESSENGER. These measurements display unusual high count rates around the sodium mass, when MESSENGER was in the solar wind at more than two Mercury radii from the surface.

Analysing the phase velocity space distribution of these counts, they concluded that these particles were non-gyrotropic ion and could be sodium ion freshly ionized and picked up by the solar wind in a region upstream to Mercury bow shock. They then estimated the size of, what they called, the interaction region, I guess between the putative cloud of neutral sodium atoms and the solar wind, by reconstructing the possible trajectory of these new ion between their position of creation of FIPS. Knowing the size of the interaction region, they then derived a rough estimate of the Na neutral density that should be at the origin of this observation.

This paper is clearly of high importance because it suggests the existence of very intense, localized and sporadic source of exospheric material at Mercury. These authors proposed that this source has been induced by the impact of a meter size object leading to an observable signature in the exosphere which seems to be a realistic explanation following theoretical past works.

The importance of this observation can be easily illustrated by a simple calculation. From their estimate of a sodium neutral density between 10^2 to 10^3 Na/cm³ at 5300 km induced by this event, and considering as the authors that these neutral particles have energy close to the escape energy, that is, a velocity of 1.4km/s at 5300 km, the neutral flux induced by this event should have been from 1.4 to 14×10^7 Na/cm²/s at 5300 km. Extrapolated back to the surface, we can then estimate that a minimum flux between 10^{25} to 10^{26} Na/s should have been ejected from the surface during this event to induce such flux at 5300 km. These numbers have to be compared to the typical rate of sodium permanently ejected from Mercury surface; at aphelion between 7.5×10^{23} Na/s to 1.5×10^{25}

Na/s according to Killen et al. (2019). In another way, this event might have induced, almost instantaneously and localized on a very small surface, the release into Mercury's exosphere of an amount of Na atoms equivalent to the total amount of sodium atoms released from the whole surface during the same time. Even if its signature into Mercury's exosphere should have lasted only few ten of minutes, it implies that an instrument observing Mercury 'exosphere would have clearly seen it.

I therefore believe that this paper deserves to be published. I have few comments that are listed below and that I consider as minor.

My first comment is related to the conclusion that the species at the origin of this unusual count rate should have been the sodium atomic species. However, as shown by Mangano et al. (2007), at least three other species within FIPS sodium group, should be also released by a meter size impact: Mg (mass 24), Al (mass 27) and Si (mass 28). For these three species, Mangano et al. (2007) predicted peak of neutral density at 1500 km larger than the peak for Na of $4 \times 10^8 \text{ m}^{-3}$: 10^{10} m^{-3} for Mg, 10^9 m^{-3} for Al and 10^{10} m^{-3} for Si. Even corrected by the observed surface composition (Peplowski et al. 2014), the expected densities at 1500 km remain very significant with respect to Na. As a matter of fact, Nittler et al. (2019) reported a Al/Si ratio 4 times larger than the one used in Mangano et al.. Moreover, as pointed out by Mangano et al. (2007), the enhancement of the density with respect to the exospheric density at 1500 km is predicted to be almost negligible for Na (see also my following comment) but by few orders of magnitude for all the other species (their table 3). At the end, Al, Mg and Si are as easily ionized as Na, since according to Fulle et al. (2007), the photo-ionization lifetime at 1 AU of Na is equal to $1.9 \times 10^5 \text{ s}$ (a value consistent with the ionization frequency given by the authors in the section "Method/Estimating neutral density"), whereas it is equal to $1.4 \times 10^3 \text{ s}$ for Al, $4.4 \times 10^4 \text{ s}$ for Si and 2.1×10^6 for Mg for quiet sun. In another way, if I should follow the arguments given by the authors to conclude that sodium atom and ion were the main source of FIPS measurement, I would actually rather conclude that Al is more probable. Clear arguments should be made by the authors to support their conclusions that sodium atoms are the main source of FIPS measurements. If not, the text should be changed accordingly.

I do not agree with sentences, line 23 and 24 "The global exosphere which extends up to 1500 km" or lines 51 and 52 "... it can be seen that the sodium exosphere does not extend far from Mercury" (as a comment of Figure 1c). This is contradictory with the definition of an exosphere: "the outermost region of a planet's atmosphere" and with the fact that an exosphere has no theoretical upper limit. Moreover, Figure 1c is misleading since it is based on an arbitrary choice of the colour scale and actually is in contradiction with Figure 1a which does show that an exosphere can be very extended. More important, figure 1c does not take into account the two scale heights structure of the exosphere as observed by Cassidy et al. (2015), see their figure 7 as an example, or as predicted by many authors (as an example by Mura et al. 2009; their figure 8). However, my comment does not contradict the key conclusion of the authors that FIPS observed indirectly an unusual increase in the neutral density. Indeed, the typical sodium neutral density expected at an altitude of 5300 km is

of the order of 1 Na/cm^3 (Mura et al. 2009), that is, two to three orders less than the neutral density needed to induce FIPS measurements according to the authors. Clearly, Figure 1c and the discussions in the second paragraph of the section “Main text” should be corrected.

I do not see the interest of Figure 1a. It does not illustrate any relation with the magnetosphere as uncorrectly written lines 42 and 43 but rather the effect of the solar radiation pressure. This observation was obtained at a True Anomaly Angle of 125° far from the aphelion and is therefore not representative of December 21st 2013.

The estimate of the neutral density from FIPS data deserves more explanation. In general, I found it not straightforward to understand how the authors derived it. I would suggest in particular to explain what is meant by “interaction region” in the section “Estimating Neutral Densities”.

Figure 1b: would it be possible to display the whole orbit of MESSENGER. Since the observation of large count (Figure 2b) lasted almost 8 mm, it might be interested to know where this observation started and when it ended. Moreover, on line 168 – 169, it is stated that “for 3 hours preceding this event the IMF was quiet”. It would be therefore useful for the reader to see where was MESSENGER during that period.

Lines 62 and 63: very few information are provided regarding the energy of the observed ions, only that the energy of the measured ions were at the limit of FIPS energy range. Does it mean that FIPS does observe only a portion of the ions produced during this event and that the derived neutral density has to be considered as a lower limit?

Line 72, line 89: -Y’ should point toward the dawn and not the dusk as written. This is actually what is written in the section “Method/Not the Foreshock”, line 294.

Figure 3 caption, 4th line, I believe that “b and c” should be “c and d”.

Line 102: “Figure 3c” should be “Figure 3d”, I believe.

Line 107: “panel c” should be “panel d”, I believe.

Lines 152 to 178: I do not understand why the main argument to reject most of the ejection mechanism is based on the capability of each mechanism to produce escaping particles. To reach an

altitude of 5300 km from Mercury surface needs a minimum energy of 1.47 eV, which is far from the 2.15 eV needed to escape Mercury. All these mechanisms can produce particles that can reach 5300 km in altitude but in a proportion which is variable. I believe that the key argument to reject one mechanisms with respect to the other is their temporal variability. Clearly, thermal desorption is not expected to change significantly on short time scale. Photon stimulated desorption might change if strong flares occur during that period as an example. Did you check if any flare events occur during that period of observation?

Line 187: I do not agree with that sentence. The typical temperature of the vaporized material during an impact has been reported to be around 4000 K (Eichhorn 1978), that is, around 0.5 eV. In another way, the vaporized particles have, for most of them, velocity below the escape velocity.

Lines 200 and 206: I do not understand the interest of this paragraph. Solar radiation pressure might limit the amount of sodium atoms able to reach 5300 km on the dayside but might help to observe such species if the main source at the surface would have been near the subsolar point (pushing the sodium atoms from noon local time towards the terminator). In the case of FIPS observation at aphelion, the trajectory of the released sodium atoms should not have been significantly changed by this effect but I think it to be a minor ingredient not essential for this observation to occur.

Reviewer #3 (Remarks to the Author):

General Comments

This paper reports the occasional observation of heavy ions at great distance from the Mercury's surface, out of the bow shock. The observation, unique in the MESSENGER data, is surely of great interest and deserves publication. The interpretation of such event will surely open new views of the Mercury's environment. Nevertheless, I think that along the paper there are some unproved conclusions that should be smoothed.

The authors interpret the observed heavy ions as recently ionized and accelerated exospheric Na released via meter-sized meteoroid impact vaporization (MIV). As first point the title is misleading. The observation is not sodium (neutral) exosphere, while it is heavy ion in a wide mass range ($m/e=21-30$, that is Na^+ , Mg^+ , Si^+ , as written in the Methods section). I would suggest a new title like: "The Mercury exosphere at very high altitudes deduced from an occasional MESSENGER/FIPS observation in the solar wind".

According to the Mangano et al 2007 paper, the probability to have a meter-size meteoroid impact is 2/year and the possibility to have particles at an initial velocity suitable to reach 5000 km and lasting for 10 minutes (FIPS integration time) is even lower. It is not clear how the authors derive from the

paper the required 10^2 Na/cm^3 density, that the authors claim as source of the observed ionized population (page 12). According to Mangano et al 2007, it is quite difficult to distinguish an enhancement due to MIV in the dayside Na exosphere. In Mangano et al 2013, I don't find any reference to the statement in the paper "Such an impact would cause strong local enhancements at large altitudes of the sodium exospheric, with neutral densities two to three orders of magnitude larger than the background exosphere required to be observed as pickup ions by FIPS". How the authors justify this sentence?

This does not mean that it is not possible at all to observe a major impact during the MESSENGER lifetime and that the observations are not related to MIV, nevertheless I would consider the whole exospheric components in the observed mass range.

Another not clear point is the justification of the energy of the ions. 13 keV is an energy that requires an acceleration mechanism, the authors neglect any in-depth study on the subject.

Finally, I think that there is some confusion in the reference frame, that is, in the dawn / dusk determination, see detailed comments below.

detailed comments

line 56: "It is very surprising that we observe Na+ in the solar wind because the average exospheric densities are only high close to the planet ...". Maybe it is better: "It is very surprising that we observe Na+ in the solar wind because generally the exospheric densities are distributed close to the planet surface..."

Fig 1a: Since, as the authors explain later, the observation refers to a different True anomaly angle, the image is not the best one. I suggest deleting it or substituting it with a ground based observation at similar TAA, i.e. close to aphelion.

In the caption I think that it is not necessary to explain the meaning of the projections.

The identification of the dawn/dusk positions is something that helps the reader. I would add in the explanation that MESSENGER was at dawn that is in the ram side of the planet, if I correctly interpret the coordinate system. In this case, the planet in figure 1c and fig 4b is seen from the night side.

This could help in the interpretation, since ram direction is the most favorable side for the meteoroid impact.

Line 59: I would avoid to write "strong magnetic field" and "very small" gyroradius, a gyroradius of $0.4 R_m$ is not so small and the magnetic field is not strong with respect to the Earth case, for example. The newly ionized ions should have almost thermal energy, that is why it is not expected to escape. The 13 keV-energy cannot be the initial ion energy.

Fig 2 b: C/acc is not the best unit. If the “accumulation” means 10 minutes, write counts/10 minutes or scale it to counts/minute.

Line 71-72 and line 89: here I am confused. The $-Y'=T$ direction should be downward as defined in the method section, not duskward.

Fig 3 caption line 4 it must be “c and d” not “b and c”

Line 127 again not very high magnetic field but higher (than in the magnetosheath)

Line 129: why the ions were observed shortly after ionization? how can they accelerate so fast? For example a shock-induced dissociative ionization from a MIV-released molecule of Mg could explain the high energy (Killen, Icarus, 2016).

Line 133: again I think that this explanation on projection is not necessary

Line 172-175: the reconnection rate at Mercury is generally high at any IMF orientation, as the authors know. So I would write: “Even if generally the reconnection rate is high at Mercury and cusp precipitation occurs also during conditions similar to the present observation, the unicity of this detection makes this case related to a different sporadic event non related to the IMF conditions.”

Lines 184: the MIV contribution to the exosphere is highly debated, it depends on the considered species and it may have asymmetries (probably higher in the ram/dawn side) so the 1% should be considered as a rough estimate.

Line 190 : in Mangano et al. 2013 the big meteoroid contribution is not discussed. The correct reference is Mangano et al 2007

Lines 191-193: this is not the outcome of the Mangano et al. 2007 paper. As written in the general comments, it is quite difficult to distinguish an enhancement due to MIV in the dayside Na exosphere.

Line 202: “radiation pressure from the Sun is at its the lowest value”

Lines 203-204: as written in the general comments, it is not useful to include a figure showing a totally different condition of the Na distribution. I would delete the image also considering that we are not sure that the observed species is Na that has a peculiar behavior (tail formation).

Line 245 “steradian”

Line 254: this conclusion is not supported and, probably, not necessary.

Lines 265-267: “These events will be investigated in the future and are largely expected to occur due to either bow shock acceleration processes (such as foreshock acceleration) or solar wind sputtering.” The interpretation of these cases is not explained and not necessary in this context.

Response to Reviewer's for the Nature Communications paper: "A transient enhancement of Mercury's exosphere observed at extremely high altitudes in the solar wind" by Jasinski et al..

We would like to, first of all, thank the reviewers for devoting their time to review our paper, and for their encouraging comments and useful suggestions. We have responded in bold to the reviewer's comments below.

The main change has been to the (previous) assumption that the observation is of Na⁺. We agree with all the reviewers that these observations may not be only Na⁺. We have therefore changed the language throughout the paper to be clear that it is not necessarily Na⁺, but ions in the "sodium-group" (or similar terminology). Therefore, throughout the paper we use the terms "heavy ions" (or heavy ions of planetary origin), "Na⁺-group ions" or more conveniently when the text discusses the pickup process - "pickup ions". We also specify this mass-per-charge binning directly in the text, and make every effort to be explicit, for example with the new paragraph (starting on Line 85):

"During the MESSENGER mission, FIPS' measurements of heavy ions were binned to increase the signal-to-noise ratio. Na⁺ measurements were binned with other species with similar mass-to-charge ratios (m/e) in what is called the sodium-group (m/e = 21–30 amu/e, including Na⁺, Mg⁺, Al⁺ and Si⁺). Therefore, it is not possible to directly distinguish between these different species, and any mention of sodium-group ions (Na⁺-group), heavy-ions or pickup ions in this paper therefore refer to the above-mentioned group of species."

The title also has now been changed to reflect this (the title does not include the word "sodium exosphere" just "exosphere"). The paper is now called "A transient enhancement of Mercury's exosphere observed at extremely high altitudes in the solar wind".

Figure 1a was deleted. Figure 1b, 1c are now Figure 1a, 1b.

Figure 1a (formerly 1b) has been shown in a new orientation. It was previously differently orientated to match the orientation of the deleted figure. It now matches the orientations of Figure 4a and 4c

Reviewer #1 (Remarks to the Author):

Summary

The observation of enhanced Na⁺ ions (really an enhancement of the "Na-group" ions – see below) at high altitude near the bow shock is indeed an interesting measurement. This can

reasonably be viewed as coming from an impact event. However, the authors are too focused on Na in their explanation and they fail to consider both alternative sources (Mg) as well as to speculate on the timely nature of a potential impactor from comet Encke. As such, the paper needs some reworking in order to be published.

Major comments

The grouping of FIPS ions (lines 249-254) points to a particularly interesting possibility not considered by the authors – that this observation pertains to Mg and not to Na. Although it is true that Na is the dominant neutral, Mg has a lifetime against photoionization that is an order of magnitude longer than Na and thus can reach higher altitudes before experiencing photoionization. The temperatures necessary for the Mg distribution observed by MESSENGER (Merkel et al. 2017) already suggest that micrometeoroid impact vaporization is the likely primary source of the Mg exosphere. Column densities of Mg (Merkel et al. 2017) compared to Na (Cassidy et al. 2015) show Na is about 10 times that of Mg, so if the Na source rate from impacts is < 20% of the total Na, Mg is an equally plausible atom. Thus, an impact may have generated a cloud of Mg rather than Na (or more likely a cloud of both). It is premature of the authors to assume this is just Na⁺ rather than Mg⁺ or some combination of the two given the limitations of the FIPS data. This discrepancy cannot be ignored, so the paper is really about the observation of a potential impact event and not strictly about the Na exosphere. I suggest a title change to that effect, and I suggest that all references to Na throughout be changed to Na-group data or Na-Mg data or something of the sort. This reworking does not diminish the observational data, nor the fact that this is an interesting observation, but simply posits an explanation that is more likely and truthful to the situation. [Note that in the remaining comments, I refer to "Na" but I am speaking of the "Na-group ions". The language is just easier to type.]

We agree and have changed the text to reflect the likelihood that the FIPS signal is a mixture of Si⁺ and Na⁺ alone. Our estimates suggest that Mg⁺ is not likely to be the main contributor (we explain this in the paragraph below). We have also changed the language throughout the paper to make sure that we do not assume it to be sodium.

We have added a section in the Methods called "Composition Estimation" which can be found from Line 427 onwards. This also includes a Table (1) to highlight the differences between the different species and why we estimate that the composition is some combination of Na and Si.

In the main text this is also summarized on Lines 266-273, which read:

"Finally, we estimate that the composition of the pickup ions is a combination of Na and Si (please see the Methods section for more details). Al has the highest ionization frequency of 10^{-3} s^{-1} (i.e. lowest photoionization lifetime of 170 s) and so the neutrals would mostly have been ionized before reaching high altitudes. Mg has a very low ionization frequency (10^{-7} s^{-1} , or a photoionization lifetime of 10^6 s) and therefore not enough ions would be produced to

provide the observed ion flux. Therefore, Si and Na are the most likely candidates to have produced the observed pickup ions.”

The paper also neglects to consider the timing of this event. Mercury had a particularly close passage to comet Encke in 2013 only 3-4 weeks before this observation. Based on simulations from Christou et al. (2015), the Encke stream not only crosses the orbit of Mercury near TAA ~ 25 degrees (where an increase in the Ca exosphere is observed), but it also crosses near TAA 130-160 degrees. This is not too far from the TAA ~ 177 degrees at which the observation occurred. The event also appears to happen on the dawn side of the planet (line 294), which is the ram direction of Mercury as it plows through the Encke stream. Because the Christou et al. work was primarily focused on smaller grains that are likely behind the Ca enhancement at Mercury, it is possible that the larger bodies needed for such an impact event as noted here would evolve under dynamic effects to slightly different TAAs. Either way, the timing may not be entirely coincidental and should be noted.

We thank the reviewer for this comment. We agree that this should be mentioned even though an asteroid source is more likely to be the cause of a single larger impact. We have therefore noted this as a possibility as the reviewer suggested, on Lines [253-256]:

“One possible source for an impact could be from the crossing of comet Encke’s stream (TAA \sim 130-160 $^\circ$; Christou et al., 2015), which Mercury completed crossing a week earlier, however an asteroid source is more likely to produce a single large impact.”

Specific comments

Abstract

The characterization of the exosphere as “closely bound” is a little misleading (line 18). The primary source of Na, photon-stimulated desorption (PSD), does indeed lead to the low-altitude Na exosphere having a small scale height. However, all the observed Na profiles extend to much higher altitudes (duly noted by the authors), with this higher-altitude component suspected of deriving from a higher energy source than PSD such as sputtering or micrometeoroid impact vaporization as only they have the energy to eject Na atoms with the velocities needed to reach these altitudes. The altitudes reached are a significant fraction of the planetary radius, so the Na exosphere is not quite as “closely bound” as the initial statement would indicate. This also completely neglects the fact that the Na tail (mentioned in the main text) is an extension of the Na exosphere that can extend many planetary radii antisunward. Reworking the opening statements is needed to be faithful to what is really the case. It does not detract from the point that neutral Na at 5300 km would be significantly less dense under normal circumstances.

We thank the reviewer for spotting this misleading text. We have edited the text to be more accurate in representing the situation. We have focused on mentioning the example densities at the dayside as a comparison of how such high densities at our high altitudes are not typical of the typical exosphere. The text now reads:

Line 38-40: “This sodium exosphere has small dayside scale heights of up to ~100 km at perihelion, with subsolar densities of 10^3 cm^{-3} measured at altitudes of ~450km [Cassidy et al., 2015].”

And on Line 18-19: “Mercury has a global dayside exosphere, which contains sodium densities of 10^{-2} cm^{-3} at ~1500 km”

The description of these as the first “measured enhancement of neutral sodium densities” (line 20) is incorrect. FIPS did NOT measure neutrals, so at best these are the first observations that INFER or SUGGEST an enhancement of neutral sodium densities.

We have changed the wording from “measured” to “inferred” as suggested (Line 19).

The statement that “no observations attributable to micrometeoroids were detected” (lines 33-34) is not true. There are no published results connecting the MESSENGER *sodium* data to micrometeoroids (although the published densities at higher altitudes suggest that as a source), but there have been several papers detailing the connections of calcium to micrometeoroids. The blanket statement in the abstract should be more representative of the situation.

We have now deleted this statement.

Main text

Line 43: The statement that the exosphere is “constrained within” the magnetosphere suggests that the magnetosphere is limiting the extent of the exosphere. I suggest that this be reworded such that the bulk of the Na exosphere simply lies within the magnetosphere.

We have changed the text now to read: “Therefore, the exosphere mostly lies within the magnetosphere of Mercury.” on Lines 44-45.

Figure 1 caption: Under the description of panel (b) it says the “View from dusk”. If the Sun is to the right, then this is the view from *dawn* unless the planet is upside down (i.e., north is down). I cannot specifically say how the authors defined the MSO system to know definitively

which way positive Z points, but I would suspect +Z is in the north direction and thus “up”. Note that I am only claiming this view is from dawn and not that any of the subsequent descriptions related to dusk in the text are incorrect.

The reviewer is correct. The view was from dawn. This was left over from a previous draft where the orientation was different. However, the new figure is now from dusk.

Figure 1, panel (c): Either capitalize “neutral” or don’t capitalize “Density”.

Done.

Line 73: The spacecraft location in Figure 1b is inside the bow shock whereas it says here that the spacecraft crossed the bow shock and entered the the magnetosheath. I realize this is probably the difference between a model bow shock in Figure 1b and reality, but the disconnect is disconcerting and I recommend some rewording to make things clearer.

This is a projection effect. We have added text to clarify on Lines 54-59:

“The bow shock in panel a is in the noon-midnight meridian (i.e. at $Y=0$), however the spacecraft is positioned downward of Mercury (i.e. $Y \sim -2 R_M$). where the bow shock is northward of MESSENGER (therefore the appearance of the spacecraft inwards of the bow shock is a projection effect, and MESSENGER lies outside the bow shock in the solar wind as can more accurately be visualized in panel b).”

Figure 3 caption: For the uninitiated, it would be useful to define what wT' , wN' and wR' are here; at the very least point them to the “Methods” section where they are defined.

We have added text in the caption: “The data are shown in aberrated RTN coordinates (R is directed in the solar wind flow direction, T is Mercury’s orbital direction and N is positive northward with respect to Mercury) which are described in detail in the Methods section.”

We have also added text on lines 116-119: “ R' is in the solar wind velocity direction and points towards the planet (and is in the $-X'_{MSO}$ direction). T' is the cross product of the Sun’s spin angular velocity vector and the R vector (i.e. T' is directed along $-Y'_{MSO}$), and N completes the right hand set (N points along Z_{MSO}).”

Line 110: I suggest “highly active comets” rather than “more active comets”; Mercury is not a comet.

We have changed the text to the suggested “highly active comets”.

Lines 150-151: Awkward sentence; rephrase.

Rephrased Lines 209: “To explain this observations, requires a **non-typical** interpretation.”

Line 153: Hyphen needed in “photon stimulated”.

Done.

Line 155: “Due to” should be “owing to”.

Changed.

Line 165: “Due to” should be “owing to”.

Changed to “At Mercury, sputtering is due to...” (229)

Lines 182-183: An example is given for the Moon, but there is ample published evidence that micrometeoroids at Mercury are liberating Ca, particularly during the passage of the comet Encke dust stream across Mercury’s orbit (Burger et al. 2014, Killen and Hahn 2015, Christou et al 2015) where the source rate enhancement is quite large. It is also likely that the process is contributing to the Na exosphere at Mercury but the signal is smaller compared to the dominant PSD source rate. Nevertheless, there is no need to invoke the Moon here.

We appreciate the reviewers reference to the work completed in regards to Mercury, however we wish to reference a body other than Mercury, as well as reference observations of larger impactors.

Lines 183-184: The numbers from Cremonese et al. (2005) and Burger et al. (2010) should be noted as pertaining specifically to a souce of the Na exosphere. As just noted, the contribution of impacts to the Ca exosphere is both dominant and large. One cannot think of Mercury as having a single exosphere anymore.

We have added “sodium exosphere” to specify this now.

Line 188: “micrometeorites” should be “micrometeoroids”.

Changed.

Line 188: “Due to” should be “owing to”.

Changed.

Line 192: “exospheric” should be “exosphere”.

Changed.

Line 196-197: It should be noted that if this Na comes from the southern hemisphere, the “measurements” of Na in Peplowski et al. (2014) are not true measurements but simply an assumption that the south is like the north owing to symmetry. However, that conclusion is based on the assumption that Na “migrates” to the poles; it is still possible that the hemispheres are just different compositionally. BepiColombo will hopefully shed light on this. Regardless, MESSENGER was too far from the southern hemisphere for the surface content of Na to be measured there. Thus, the order of magnitude difference cited relative to Mangano et al. (2007) may not be correct and that caveat should be noted.

We have added this caveat, now mentioned on Line 263: “....and assuming that the surface composition in the southern hemisphere is similar (to an order of magnitude) to the north...”

Lines 197-199: Awkward sentence with badly placed commas; please rephrase.

This sentence has been broken up into two sentences now.

Lines 200-206: The tail extension occurs at two points in Mercury’s orbit, not just one (it’s just that because of the motion of Mercury the tail is bigger/stronger on one part than the other). It is true that the tail at aphelion is small to nonexistent, so the point about trajectories being less altered is correct. That’s all that needs to be said, but if the point about the tail is to be kept, it needs to be restated to be factually correct.

We have now removed this paragraph.

Line 253: The “[Killen et al., 2018]” reference should be “[Killen et al., 2019]”.

This sentence has been deleted now.

Line 304: “Since” should be “Because”.

Done.

Reviewer #2 (Remarks to the Author):

This paper reports original measurements performed by the Fast Imaging Plasma Spectrometer (FIPS) on board MESSENGER. These measurements display unusual high count rates around the

sodium mass, when MESSENGER was in the solar wind at more than two Mercury radii from the surface.

Analysing the phase velocity space distribution of these counts, they concluded that these particles were non-gyrotropic ion and could be sodium ion freshly ionized and picked up by the solar wind in a region upstream to Mercury bow shock. They then estimated the size of, what they called, the interaction region, I guess between the putative cloud of neutral sodium atoms and the solar wind, by reconstructing the possible trajectory of these new ion between their position of creation of FIPS. Knowing the size of the interaction region, they then derived a rough estimate of the Na neutral density that should be at the origin of this observation.

This paper is clearly of high importance because it suggests the existence of very intense, localized and sporadic source of exospheric material at Mercury. These authors proposed that this source has been induced by the impact of a meter size object leading to an observable signature in the exosphere which seems to be a realistic explanation following theoretical past works.

The importance of this observation can be easily illustrated by a simple calculation. From their estimate of a sodium neutral density between 10^2 to 10^3 Na/cm³ at 5300 km induced by this event, and considering as the authors that these neutral particles have energy close to the escape energy, that is, a velocity of 1.4km/s at 5300 km, the neutral flux induced by this event should have been from 1.4 to 14×10^7 Na/cm²/s at 5300 km. Extrapolated back to the surface, we can then estimate that a minimum flux between 10^{25} to 10^{26} Na/s should have been ejected from the surface during this event to induce such flux at 5300 km. These numbers have to be compared to the typical rate of sodium permanently ejected from Mercury surface; at aphelion between 7.5×10^{23} Na/s to 1.5×10^{25} Na/s according to Killen et al. (2019). In another way, this event might have induced, almost instantaneously and localized on a very small surface, the release into Mercury's exosphere of an amount of Na atoms equivalent to the total amount of sodium atoms released from the whole surface during the same time. Even if its signature into Mercury's exosphere should have lasted only few ten of minutes, it implies that an instrument observing Mercury 'exosphere would have clearly seen it.

I therefore believe that this paper deserves to be published. I have few comments that are listed below and that I consider as minor.

We thank the reviewer for their positive and encouraging comments.

#1

My first comment is related to the conclusion that the species at the origin of this unusual count rate should have been the sodium atomic species. However, as shown by Mangano et al. (2007), at least three other species within FIPS sodium group, should be also released by a meter size impact: Mg (mass 24), Al (mass 27) and Si (mass 28). For these three species, Mangano et al. (2007) predicted peak of neutral density at 1500 km larger than the peak for Na of 4×10^8 m⁻³:

1010 m⁻³ for Mg, 109 m⁻³ for Al and 1010 m⁻³ for Si. Even corrected by the observed surface composition (Peplowski et al. 2014), the expected densities at 1500 km remain very significant with respect to Na. As a matter of fact, Nittler et al. (2019) reported a Al/Si ratio 4 times larger than the one used in Mangano et al.. Moreover, as pointed out by Mangano et al. (2007), the enhancement of the density with respect to the exospheric density at 1500 km is predicted to be almost negligible for Na (see also my following comment) but by few orders of magnitude for all the other species (their table 3). At the end, Al, Mg and Si are as easily ionized as Na, since according to Fulle et al. (2007), the photo-ionization lifetime at 1 AU of Na is equal to 1.9×10⁵ s (a value consistent with the ionization frequency given by the authors in the section “Method/Estimating neutral density”), whereas it is equal to 1.4×10³ s for Al, 4.4×10⁴ s for Si and 2.1×10⁶ for Mg for quiet sun. In another way, if I should follow the arguments given by the authors to conclude that sodium atom and ion were the main source of FIPS measurement, I would actually rather conclude that Al is more probable. Clear arguments should be made by the authors to support their conclusions that sodium atoms are the main source of FIPS measurements. If not, the text should be changed accordingly.

We agree that Na⁺ is not likely to be the only contributor to this signal. Based on extrapolations from Mangano et al., 2007, we find that Si⁺ and Na⁺ are most likely the dominant contributors. Mg⁺ is a minor one because of its low photoionization rate. Al was another candidate in the Na-group mass range, but its photoionization rate is so high that Al ejected from the surface cannot reach the altitude where the FIPS observed the pickup ions. The Al photoionization lifetime is about 170 s at Mercury aphelion (based on <https://phidrates.space.swri.edu>), while a neutral Al atom would take 0.5-1 hr to reach 5300 km after being ejected from the surface. Only a tiny fraction of Al atoms would survive that trip.

We have changed the text to reflect the likelihood that the FIPS signal is a mixture of Si⁺ and Na⁺ alone. We have also changed the language throughout the paper to make sure that we do not assume it to be sodium.

We have added a section in the Methods called “Composition Estimation” which can be found from Line 427 onwards. This also includes a Table (1) to highlight the differences between the different species and why we estimate that the composition is some combination of Na and Si.

In the main text this is also summarized on Lines 266-273, which read:

“Finally, we estimate that the composition of the pickup ions is a combination of Na and Si (please see the Methods section for more details). Al has the highest ionization frequency of 10⁻³ s⁻¹ (i.e. lowest photoionization lifetime of 170 s) and so the neutrals would mostly have been ionized before reaching high altitudes. Mg has a very low ionization frequency (10⁻⁷ s⁻¹, or a photoionization lifetime of 10⁶ s) and therefore not enough ions would be produced to provide the observed ion flux. Therefore, Si and Na are the most likely candidates to have produced the observed pickup ions.”

#2

I do not agree with sentences, line 23 and 24 “The global exosphere which extends up to 1500 km” or lines 51 and 52 “... it can be seen that the sodium exosphere does not extend far from Mercury” (as a comment of Figure 1c). This is contradictory with the definition of an exosphere: “the outermost region of a planet’s atmosphere” and with the fact that an exosphere has no theoretical upper limit. Moreover, Figure 1c is misleading since it is based on an arbitrary choice of the colour scale and actually is in contradiction with Figure 1a which does show that an exosphere can be very extended. More important, figure 1c does not take into account the two scale heights structure of the exosphere as observed by Cassidy et al. (2015), see their figure 7 as an example, or as predicted by many authors (as an example by Mura et al. 2009; their figure 8). However, my comment does not contradict the key conclusion of the authors that FIPS observed indirectly an unusual increase in the neutral density. Indeed, the typical sodium neutral density expected at an altitude of 5300 km is of the order of 1 Na/cm^3 (Mura et al. 2009), that is, two to three orders less than the neutral density needed to induce FIPS measurements according to the authors. Clearly, Figure 1c and the discussions in the second paragraph of the section “Main text” should be corrected.

To be more accurate we have changed the first two sentences (that the reviewer mentioned) to:

Lines (18) “Mercury has a global **dayside** exosphere, **which contains sodium densities of 10^2 cm^{-3}** at $\sim 1500 \text{ km}$ ”

Lines(61-63) “On the right (**panel b**), it can be seen that **high densities ($\sim 10^2 \text{ cm}^{-3}$)** of the sodium exosphere **are located close to the planet**, and with a small scale height the neutral density is extremely tenuous at altitudes greater than **2000 km on the dayside**”

Figure 1a has also now been deleted (at the suggestion of yourself and other reviewers).

We believe that Figure 1b (previously 1c) is not misleading as it displays how the dense regions ($10^2 - 10^5 \text{ cm}^{-3}$) of the sodium exosphere are expected to be close to the surface. The goal is to help the reader visualize how unexpected the estimated densities are at 5300 km altitudes where one would expect very low densities. However, we agree that there are, of course, some caveats to what is shown, so we have mentioned them in the text. In the Figure 1 caption it now reads: “This model exosphere is extremely simple and does not capture many of the details of the actual exosphere. The model is only intended to show how high neutral sodium densities ($>10^2 \text{ cm}^{-3}$) are typically expected to be close to the dayside surface in comparison to MESSENGER’s location at 5300 km.”

#3

I do not see the interest of Figure 1a. It does not illustrate any relation with the magnetosphere as uncorrectly written lines 42 and 43 but rather the effect of the solar radiation pressure. This observation was obtained at a True Anomaly Angle of 125° far from the aphelion and is therefore not representative of December 21st 2013.

This figure has now been deleted.

#4

The estimate of the neutral density from FIPS data deserves more explanation. In general, I found it not straightforward to understand how the authors derived it. I would suggest in particular to explain what is meant by “interaction region” in the section “Estimating Neutral Densities”.

We have added text to explain what we mean by interaction region.

Line 174-177: “The interaction region which we are trying to determine, is the approximate size of the neutral “cloud” or “plume” that the ions are coming from. The size of the region will affect the ion flux that is observed (discussed below).”

In the Methods we have also added text (Lines 409-425):

“Therefore, the size of the interaction region (i.e. the size of the neutral cloud where the particles are being ionized) is important, as it will affect the ion flux that we observe.

Modeling the neutral cloud as a simple sphere, box or cylinder with a radius or length of $0.5 R_M$, gives us the number of particles inside the cloud by multiplying by the volume. Multiplying by the ionization frequency then gives us the number of ions being produced per second. These ions then begin to flow in the direction of the electric field, out of the neutral cloud (anisotropically) through an area dependant on the size of the cloud, to produce an ion flux (i.e. $flux = \lambda n_{NA} v/a$). Therefore, the ion flux is dependant on both the size of the cloud and the density. By constraining the size of the cloud (from the observations and the particle tracing effort) we can estimate the density of the neutral cloud. We estimate that the sodium neutral densities required to produce the ions fluxes (that we observed) is $0.5 \times 10^3 \text{ cm}^{-3}$. With this simple function, for a fixed flux, n_{NA} is dependent on the length scale x as the $v/a \sim x$. With a larger interaction region (e.g. 1 or $2 R_M$), the required neutral density is lower $\sim 10^2 \text{ cm}^{-3}$ (for both 1 and $2 R_M$).”

#5

Figure 1b: would it be possible to display the whole orbit of MESSENGER. Since the observation of large count (Figure 2b) lasted almost 8 mm, it might be interested to know where this

observation started and when it ended. Moreover, on line 168 – 169, it is stated that “for 3 hours preceding this event the IMF was quiet”. It would be therefore useful for the reader to see where was MESSENGER during that period.

We have added a green line to the Figure to illustrate the spacecraft trajectory whilst in the solar wind.

#6

Lines 62 and 63: very few information are provided regarding the energy of the observed ions, only that the energy of the measured ions were at the limit of FIPS energy range. Does it mean that FIPS does observe only a portion of the ions produced during this event and that the derived neutral density has to be considered as a lower limit?

We have added text to the main article discussing the energy on Lines 151-169:

“The energy of the pickup ions measured by FIPS was found to have an energy distribution of 9 – 13 keV/e, which is at the upper limit of the instruments energy range (0.046 – 13 keV/e). The ions will initially have very low energies, however it is the pickup process that transfers energy from the solar wind into the newborn ions and accelerates them to the energies we observe here. This acceleration is initially completed by the solar wind motional electric field, and the maximum energy is dependent on the angle between the magnetic field and V_{SW} . At comets for a $V_{SW} = 440 \text{ km s}^{-1}$ solar wind, H_2O^+ can be accelerated up to energies of 70 keV for an angle of 90° [e.g. Coates et al., 2016]. During our event, the motional electric field is estimated to be 0.002 V/m, and this electric field is high enough to accelerate a sodium ion to the observed energies (from rest) within $\sim 30 \text{ s}$. This is much smaller than the estimated gyroperiod of ~ 5 minutes, and would mean that the ion would be expected to be observed in the lower left quadrant of Figure 3d (as not enough time has passed for it to make a full gyration), which is where it is detected. The pickup process would be expected to eventually accelerate the ions to much higher energies than observed. We are however, detecting these ions midway through this acceleration process, and therefore we do not expect that we are “missing” much of the ion distribution (due to FIPS’ restricted field of view) considering the ions are observed as a localized non-gyrotropic beam.”

Line 72, line 89: -Y’ should point toward the dawn and not the dusk as written. This is actually what is written in the section “Method/Not the Foreshock”, line 294.

Figure 3 caption, 4th line, I believe that “b and c” should be “c and d”.

Line 102: “Figure 3c” should be “Figure 3d”, I believe.

Line 107: “panel c” should be “panel d”, I believe.

We thank the reviewer for catching these mistakes. They have been corrected.

#7

Lines 152 to 178: I do not understand why the main argument to reject most of the ejection mechanism is based on the capability of each mechanism to produce escaping particles. To reach an altitude of 5300 km from Mercury surface needs a minimum energy of 1.47 eV, which is far from the 2.15 eV needed to escape Mercury. All these mechanisms can produce particles that can reach 5300 km in altitude but in a proportion which is variable. I believe that the key argument to reject one mechanisms with respect to the other is their temporal variability. Clearly, thermal desorption is not expected to change significantly on short time scale. Photon stimulated desorption might change if strong flares occur during that period as an example. Did you check if any flare events occur during that period of observation?

We agree that the energy alone does not rule out any single process. But thermally desorbed or photodesorbed atoms are much less likely to reach such high altitudes as compared to impact vaporization or sputtering. The low-energy portion of the Na exosphere, for example, is typically confined to within about 1000 km altitude

We have added this to the text on Line (224-227) “Thermally desorbed or photodesorbed atoms are much less likely to reach such high altitudes as compared to impact vaporization or sputtering. The low-energy portion of the Na exosphere, for example, is typically confined to within about 1000 km altitude.”

Regarding photon stimulated desorption as a possible source: we did not find evidence of large changes in solar irradiance during that time period. We looked into datasets on solar irradiance such as

<http://lasp.colorado.edu/lisird/data/fism>

That example is for far ultraviolet (FUV) and extreme ultraviolet (EUV). PSD is also caused by near-UV light (see Yakshinskiy and Madey, 1999, Nature, Fig. 1a), where solar variability is minimal compared to the EUV and FUV.

#8

Line 187: I do not agree with that sentence. The typical temperature of the vaporized material during an impact has been reported to be around 4000 K (Eichhorn 1978), that is, around 0.5 eV. In another way, the vaporized particles have, for most of them, velocity below the escape velocity.

Yes, we agree that the average energy of impact vapor is too low to escape. However, a significant fraction can reach the altitude of our observation. We have changed the text to clarify this, Line 251: “The impact will vaporize sodium with high enough energies that a

fraction of the atoms will reach the altitude of the FIPS observation.”

Also, on line 218 we replaced the escape energy, which is not the relevant quantity, as you pointed out, with the energy needed to reach 5300 km altitude (~1.5 eV for Na).

#9

Lines 200 and 206: I do not understand the interest of this paragraph. Solar radiation pressure might limit the amount of sodium atoms able to reach 5300 km on the dayside but might help to observe such species if the main source at the surface would have been near the subsolar point (pushing the sodium atoms from noon local time towards the terminator). In the case of FIPS observation at aphelion, the trajectory of the released sodium atoms should not have been significantly changed by this effect but I think it to be a minor ingredient not essential for this observation to occur.

We have deleted this paragraph as suggested by the reviewer.

Reviewer #3 (Remarks to the Author):

General Comments

This paper reports the occasional observation of heavy ions at great distance from the Mercury's surface, out of the bow shock. The observation, unique in the MESSENGER data, is surely of great interest and deserves publication. The interpretation of such event will surely open new views of the Mercury's environment. Nevertheless, I think that along the paper there are some unproved conclusions that should be smoothed.

#1

The authors interpret the observed heavy ions as recently ionized and accelerated exospheric Na released via meter-sized meteoroid impact vaporization (MIV). As first point the title is misleading. The observation is not sodium (neutral) exosphere, while it is heavy ion in a wide mass range ($m/e=21-30$, that is Na^+ , Mg^+ , Si^+ , as written in the Methods section). I would suggest a new like: "The Mercury exosphere at very high altitudes deduced from an occasional MESSENGER/FIPS observation in the solar wind".

We have changed the title to "A transient enhancement of Mercury's exosphere observed at extremely high altitudes in the solar wind".

#2

According to the Mangano et al 2007 paper, the probability to have a meter-size meteoroid impact is 2/year and the possibility to have particles at an initial velocity suitable to reach 5000 km and lasting for 10 minutes (FIPS integration time) is even lower.

It is not clear how the authors derive from the paper the required 10^{12} Na/cm³ density, that the authors claim as source of the observed ionized population (page 12). According to Mangano et al 2007, it is quite difficult to distinguish an enhancement due to MIV in the dayside Na exosphere. In Mangano et al 2013, I don't find any reference to the statement in the paper "Such an impact would cause strong local enhancements at large altitudes of the sodium exospheric, with neutral densities two to three orders of magnitude larger than the background exosphere required to be observed as pickup ions by FIPS". How the authors justify this sentence?

This does not mean that it is not possible at all to observe a major impact during the MESSENGER lifetime and that the observations are not related to MIV, nevertheless I would consider the whole exospheric components in the observed mass range.

We have now deleted the sentence the reviewer is referring to. The paper now includes references to other species that could cause the enhancement. Based on extrapolations from the Mangano et al., paper, Si⁺ and Na⁺ are the most likely contributors. As the reviewer wrote, Mangano et al., concludes that the impact is difficult to discern against the background Na exosphere. But that calculation was done for much lower altitudes than the FIPS observation at 5300 km. Energetic sodium from an impact will stand out at such high altitudes, where the background exosphere density is extremely low.

#3

Another not clear point is the justification of the energy of the ions. 13 keV is an energy that requires an acceleration mechanism, the authors neglect any in-depth study on the subject. Finally, I think that there is some confusion in the reference frame, that is, in the dawn / dusk determination, see detailed comments below.

We have added text to the main article discussing the energy on Lines 151-169:

"The energy of the pickup ions measured by FIPS was found to have an energy distribution of 9 – 13 keV/e, which is at the upper limit of the instruments energy range (0.046 – 13 keV/e). The ions will initially have very low energies, however it is the pickup process that transfers energy from the solar wind into the newborn ions and accelerates them to the energies we observe here. This acceleration is initially completed by the solar wind motional electric field, and the maximum energy is dependent on the angle between the magnetic field and V_{sw} . At comets for a $V_{sw} = 440$ km s⁻¹ solar wind, H₂O⁺ can be accelerated up to energies of 70 keV for an angle of 90° [e.g. Coates et al., 2016]. During our event, the motional electric field is estimated to be 0.002 V/m, and this electric field is high enough to accelerate a sodium ion to

the observed energies (from rest) within ~30 s. This is much smaller than the estimated gyroperiod of ~5 minutes, and would mean that the ion would be expected to be observed in the lower left quadrant of Figure 3d (as not enough time has passed for it to make a full gyration), which is where it is detected. The pickup process would be expected to eventually accelerate the ions to much higher energies than observed. We are however, detecting these ions midway through this acceleration process, and therefore we do not expect that we are “missing” much of the ion distribution (due to FIPS’ restricted field of view) considering the ions are observed as a localized non-gyrotropic beam.”

detailed comments

line 56: “It is very surprising that we observe Na⁺ in the solar wind because the average exospheric densities are only high close to the planet ...”. Maybe it is better: “It is very surprising that we observe Na⁺ in the solar wind because generally the exospheric densities are distributed close to the planet surface...”

This has now been changed as suggested.

#4

Fig 1a: Since, as the authors explain later, the observation refers to a different True anomaly angle, the image is not the best one. I suggest deleting it or substituting it with a ground based observation at similar TAA, i.e. close to aphelion.

Figure 1a has now been deleted as suggested by other reviewers.

In the caption I think that it is not necessary to explain the meaning of the projections. The identification of the dawn/dusk positions is something that helps the reader. I would add in the explanation that MESSENGER was at dawn that is in the ram side of the planet, if I correctly interpret the coordinate system. In this case, the planet in figure 1c and fig 4b is seen from the night side.

This could help in the interpretation, since ram direction is the most favorable side for the meteoroid impact.

Figure 1c (now 1b) and 4b are “seen” from the Sun (not from the nightside).

Line 59: I would avoid to write “strong magnetic field” and “very small” gyroradius, a gyroradius of 0.4 R_m is not so small and the magnetic field is not strong with respect to the Earth case, for example. The newly ionized ions should have almost thermal energy, that is why it is not expected to escape. The 13 keV-energy cannot be the initial ion energy.

This has been changed - Line(69-70): “due to the stronger magnetic fields (close to the planet). Stronger planetary magnetic field magnitudes will keep a heavy planetary ions’ gyroradius smaller”.

This sentence is showing that at these energies, the gyroradius is much different depending on the local magnetic field strength; and that even at the observed energies, the high magnetic fields close to the planet would act to trap the sodium ions inside the magnetosphere. Therefore, we do not expect these ions to simply have “gyrorated” out to the solar wind – and therefore have be “born” in the solar wind. We have also addressed the question about the acceleration process above (comment #3).

Fig 2 b: C/acc is not the best unit. If the “accumulation” means 10 minutes, write counts/10 minutes or scale it to counts/minute.

This has been changed to C/s now

Line 71-72 and line 89: here I am confused. The $-Y'=T$ direction should be dawnward as defined in the method section, not duskward.

These have been corrected to read “dawnward”.

Fig 3 caption line 4 it must be “c and d” not “b and c”
This has been corrected as suggested.

Line 127 again not very high magnetic field but higher (than in the magnetosheath)
This has been changed to “higher”.

#5

Line 129: why the ions were observed shortly after ionization? how can they accelerate so fast? For example a shock-induced dissociative ionization from a MIV-released molecule of Mg could explain the high energy (Killen, Icarus, 2016).

The energization has been addressed in the text above in comment #3.

Line 133: again I think that this explanation on projection is not necessary

This was not clear to another reviewer so we have decided to keep it in.

Line 172-175: the reconnection rate at Mercury is generally high at any IMF orientation, as the authors know. So I would write: “Even if generally the reconnection rate is high at Mercury and cusp precipitation occurs also during conditions similar to the present observation, the unicity of this detection makes this case related to a different sporadic event non related to the IMF conditions.”

We agree with the reviewer that the reconnection rate is generally high at any IMF orientation at Mercury in comparison to other planets. But that does not mean that the

reconnection rate and the effect on magnetospheric driving and particle precipitation in the cusp does not vary for different IMF conditions at Mercury [e.g. Jasinski et al., 2017]. We do not wish to imply that extreme sputtering is occurring at all times at Mercury, which is what this statement may imply considering the quiet IMF conditions during this event. The IMF conditions during this event were very “quiet” and one would not expect anything out of the ordinary in regards to reconnection, which is what we have stated in the text.

Lines 184: the MIV contribution to the exosphere is highly debated, it depends on the considered species and it may have asymmetries (probably higher in the ram/dawn side) so the 1% should be considered as a rough estimate.

We have edited the text to account for this comment. The sentence now reads (Lines 249): “...impact vaporization is **highly debated**, and is estimated to account from as little as 1% of the total contribution to the exosphere...”

Line 190 : in Mangano et al. 2013 the big meteoroid contribution is not discussed. The correct reference is Mangano et al 2007

Changed to 2007.

Lines 191-193: this is not the outcome of the Mangano et al. 2007 paper. As written in the general comments, it is quite difficult to distinguish an enhancement due to MIV in the dayside Na exosphere.

We have now deleted this statement.

Line 202: “radiation pressure from the Sun is at its the lowest value”

This statement has been deleted due to the suggestion of a different reviewer.

Lines 203-204: as written in the general comments, it is not useful to include a figure showing a totally different condition of the Na distribution. I would delete the image also considering that we are not sure that the observed species is Na that has a peculiar behavior (tail formation).

This figure has been deleted.

Line 245 “steradian”

Changed.

Line 254: this conclusion is not supported and, probably, not necessary.

Deleted.

Lines 265-267: “These events will be investigated in the future and are largely expected to occur due to either bow shock acceleration processes (such as foreshock acceleration) or solar wind sputtering.” The interpretation of these cases is not explained and not necessary in this context.

This sentence has now been deleted.

**Best Regards,
Jamie M. Jasinski (on behalf of all the authors).**

REVIEWER COMMENTS

Reviewer #1 (Remarks to the Author):

The authors have done a nice job of reworking the paper along the initial reviews; that is greatly appreciated. However, there are still some issues that require revision. This is an excellent result, but the context needs to be clear for the results to warrant publication in Nature.

Major Comments

The Abstract and Introduction are still Na-centric. I recommend that these be changed to be more general. It is entirely doable to discuss enhanced densities over typical exospheric values without specifying numbers for a given species. The simple fact that FIPS did not see anything like this over four years suggests that densities are higher than normal.

There are other instances where this still occurs as well, notably:

Lines 78-84: "Sodium-group" is introduced before being defined. Simply referring to ions with mass per charge around Na and then giving Na as an example to show these must be planetary ions is acceptable. Then you can say why it is not possible to nail the species down and define the group.

Lines 161-162: The presentation is focused again on sodium, so the text should be changed to reflect a range over the different possible ions rather than just a sodium number (unless ~ 30 s and ~ 5 minutes are good enough for all of them, and you can state that).

Lines 185-187: Similar to previous comment, a range of numbers for the possible ions unless they are all close enough to the same estimate.

Lines 199-201: Another instance of possibly needing a range of values rather than just a Na number.

Lines 204-210: This paragraph is still holding onto the idea that Na is behind all this, when it could be other species. The required neutral densities would be different for each of the species these pickup

ions could be, because of the variations in ionization rate, so citing the density required at a given altitude is different for each species. Greater credence to the assumptions of these being Na⁺ and Si⁺ would be gained by demonstrating that neither Mg nor Al results in believable densities at that altitude. Regardless of how this is handled, this paragraph needs to be made more general for the group of possible ions. This is touched upon in the Methods section, though more needs to be done there to describe the results (see additional comments below).

I realize that the above bits of text were generated because of the original focus on Na, but it would be little effort to rework these few sections to be more representative of the true situation without having to redo any significant reanalysis.

Lines 266-273: The arguments for Na⁺ and Si⁺ dominating are based on the assumption of *atoms* being released from the surface. There is a growing body of evidence that impacts can release molecules that subsequently photodissociate to the component atoms. It is beyond the scope of this paper to delve into this issue; however, it is worth a sentence or two to note that it happens. I suggest something like this right after the point about the Al photoionization rate:

“However, we note that there is growing evidence for the possibility that micrometeoroid impacts release molecules that subsequently photodissociate into the neutral atoms that are observed at higher altitudes (e.g., for the Ca exosphere, Killen and Hahn, 2015). Should such a process apply to Al, this species could also comprise some fraction of the observed heavy ions.”

I note that in lines 452-459, this argument is already mentioned in a different context for Si. If the argument can be applied to limit Si, it can also be applied to enhance Al, so the text here and in the supplemental section should be changed to qualitatively reflect the possibility of an Al contribution. I agree that Mg is less likely to be a constituent as the dominant factor for Mg is the long photoionization time.

General Comments

I suggest replacing all references in the main text to a figure “panel x” with “Figure Nx” so that it is clear which figure is being referenced. Although some instances are close enough to the original figure reference to be reasonably clear, not all occurrences throughout the paper are.

In the figure captions, the use of a single) with the panels is awkward in appearance. I suggest using two, for example (a) instead of a), in all figure captions.

Specific Comments

Line 18: Suggest changing “which contains” to “with measured”.

Line 25: Need a space between 700 and km.

Line 40: Need a space between 450 and km.

Line 48: Replace the “st” after 21 with a comma.

Figure 2 caption: Replace the “st” after 21 with a comma.

Figure 3 caption: References to “sodium ions” need to be changed to “sodium-group ions”. There are three of them.

Line 145: Add “26P/” in front of “Grigg”. Although there may not be another comet Grigg-Skjellerup, this is the standard name of this comet.

Line 152: Change “instruments” to “instrument’s”.

Figure 4 caption: Change “extended sodium exosphere event” to “transient exosphere event”.

Figure 4: The idea that these heavy ions are mostly Na⁺ and Si⁺ has not been introduced yet, so I suggest that the “Na” and “Si” in the figure be replaced with “Neutral atom” and the “Na⁺” and “Si⁺” with “Ion”, and the caption changed accordingly. This will work for whatever the composition of the heavy ions ends up being.

Line 206: Change “methods” to “Methods”.

Line 213: Delete “sodium” to keep this general to all species.

Line 218: This number is for Na (presumably), but I suspect it is close enough to say that it applies to all the sodium-group ions. The point is that thermal desorption is not a player here, and that will hold for all of the possible species.

Line 220: Change “the sodium” to “an atom” to keep this general.

Lines 225-227: It is fine to keep this as written, because Na is just given as an example.

Line 251: Before the sentence beginning “The impact...”, I suggest adding the following.

“In contrast, impact vaporization has been postulated as the dominant source of both the Ca and Mg exospheres at Mercury [Burger et al., 2014; Merkel et al., 2016].”

Lines 251-253: Change the beginning of the sentence “The impact...” to be

“In any case, an impact will vaporize the sodium-group species with high enough...”.

This will keep the argument general.

Line 258: Need a space between 1.5 and m.

Line 260: Change “exospheric” to “sodium exospheric”.

Line 260: Change “1m meteorite” to “1-m meteoroid”.

Line 265: Change “neutral densities” to “neutral sodium densities”.

Lines 277-278: The reference to Figure 4c (the first for that panel, I think) is far enough away from the Figure 4 discussion and is more qualitative than the rest of Figure 4 that this could conceivably be a Figure 5 and located nearer to this part of the text.

Line 318: Need to add Al⁺ to the list of Na⁺-group ions.

Line 367: Change “sodium” to “sodium-group”.

Line 392: Change “sodium” to “sodium-group”.

Lines 398-401: Suggest moving these lines to the end of the section (lines 421-425) and keeping this section general to any species. At the end of the section, these can be combined with lines 421-425 to provide an example for sodium.

Reviewer #2 (Remarks to the Author):

The authors answer properly to my comments and changed the paper accordingly. I therefore recommend this paper for publication. I have only few very minor comments which do not need a new review:

Line 37: I would replace “most abundant species” by “most abundant observed species”. There is a strong bias associated with the detection of Na atoms when using visible spectrometer. Therefore, I would remain careful regarding the real composition of Mercury’s exosphere.

Line 44: I do not understand the logic of this "Therefore". There is no relation between the sodium radiation pressure and Mercury's magnetosphere and therefore no reason to associate the two as it seems to be done.

Line 79-80: is this sentence related to sodium species needed here? I would suppress it to avoid any confusion regarding the sodium-group composition.

Line 83: same remark: I would replace "sodium" by "sodium-group"

Line 209: same remark than for Line 83.

Line 213: I would suppress "sodium", since the discussion regarding the exosphere should be extended to other species. As a matter of fact, this whole paragraph (up to line 273) is really focused on the origin of the sodium in the exosphere. Since, the main conclusion is that the detection might be also due to other species, would it be not more relevant to change slightly this part by providing some information regarding the origins of the other exospheric species? As an example, lines 266-267 might surprise the reader since Al, Si and Mg are not discussed at all previously.

Line 455: Killen et al. (2005) explained the origin of the very large height scale of the exospheric Ca by suggesting that Ca was ejected as CaO and then dissociated by photo-dissociation providing energy to the dissociated component. Therefore, if Si is preferentially ejected as SiO a similar process could happen and actually favour the capacity of Si to reach 5300 km altitude.

Reviewer #3 (Remarks to the Author):

The manuscript has greatly improved over the previous version. I still have problem with the title, it is not an exospheric observation. It is an ion measurement that infers the exosphere, so I would substitute the word "observed" with "inferred", as in the abstract and as suggested by reviewer 1.

As I wrote in the previous review, the observation, unique in the MESSENGER data, is surely of great interest and deserves publication. The interpretation of such event will surely open new views of the Mercury's environment. Finally, I recommend the publication after the title change.

Response to Reviewers for the Nature Communications paper: “A transient enhancement of Mercury’s exosphere at extremely high altitudes inferred from pickup ions” - by Jasinski et al.

We thank all three reviewers for devoting their time to review our corrections. We have responded in bold to the reviewer’s comments below.

We have also changed the title to use the word “inferred from pickup ions” (rather than observed in the solar wind). We have also made what was formerly Figure 4c, into a separate figure, Figure 5 (as suggested by a reviewer). This figure has also been updated only in terms of the quality of the illustration (it has not changed in terms of content).

**Best Regards,
Jamie M. Jasinski (on behalf of all the authors).**

Reviewer #1 (Remarks to the Author):

The authors have done a nice job of reworking the paper along the initial reviews; that is greatly appreciated. However, there are still some issues that require revision. This is an excellent result, but the context needs to be clear for the results to warrant publication in Nature.

Major Comments

The Abstract and Introduction are still Na-centric. I recommend that these be changed to be more general. It is entirely doable to discuss enhanced densities over typical exospheric values without specifying numbers for a given species. The simple fact that FIPS did not see anything like this over four years suggests that densities are higher than normal.

We have deleted references to a ‘sodium’ exosphere – and just discuss the exosphere. We still mention densities, so we can compare our observation estimates, but we do not mention a particular species.

There are other instances where this still occurs as well, notably:

Lines 78-84: “Sodium-group” is introduced before being defined. Simply referring to ions with mass per charge around Na and then giving Na as an example to show these must be planetary ions is acceptable. Then you can say why it is not possible to nail the species down and define the group.

We have now changed the order of the text as suggested. The text now discusses that ions within a mass-to-charge range (21-30) were detected, and that sodium lies within this range and is a major constituent of the exosphere. Then the text moves on to discuss the binning of FIPS and then defines “sodium-group”. (Lines 76-91; the text was largely rearranged which hasn’t been bolded, new text has been bolded.)

“A timeseries of the MESSENGER plasma and magnetic field observations can be seen in Figure 2. Background-level proton fluxes were measured (**Figure 2a**) in the solar wind. High counts of **heavy ions (Figure 2b) with a mass-to-charge (m/e) ratio of 21 – 30 amu/e** were observed (starting at 00:26 UT) with energies centered at 10 keV/e (close to the limit of the FIPS energy range of ~13 keV/e). Sodium (**which lies within this m/e range**) is a major constituent of Mercury's exosphere and is easily photoionized to Na⁺. In contrast, Na⁺ in the solar wind are very rare, and, if present, would very likely be detected in a higher charge state. Therefore, these heavy ions are of planetary origin.

During the MESSENGER mission, FIPS’ measurements of heavy ions were binned to increase the signal-to-noise ratio. Na⁺ measurements were binned with other species with similar mass-to-charge ratios in what is called the “sodium-group” ($m/e = 21\text{--}30$ amu/e, including Na⁺, Mg⁺, Al⁺ and Si⁺). Therefore, it is not possible to directly distinguish between these different species, and any mention of sodium-group ions (Na⁺-group), heavy-ions or pickup ions in this paper therefore refer to the above-mentioned group of species.”

Lines 161-162: The presentation is focused again on sodium, so the text should be changed to reflect a range over the different possible ions rather than just a sodium number (unless ~30 s and ~5 minutes are good enough for all of them, and you can state that).

We have updated the text as suggested (i.e. it is not focused on sodium); ~30 s is the same for all of the species , but the gyroperiod ranges from 5-6 minutes. (Lines 162-163):

“...within ~30 s (from rest **up to 10 keV/e for Na⁺, Mg⁺, Al⁺ and Si⁺**). This is much smaller than the estimated gyroperiod of 5-6 minutes (**for this group of species**)”

Lines 185-187: Similar to previous comment, a range of numbers for the possible ions unless they are all close enough to the same estimate.

We have edited this as suggested (Line 188): “...5.6 – 6.2 R_M (for a pickup ion in a 5 nT magnetic field that is either Na⁺, Mg⁺, Al⁺, or Si⁺).”

Lines 199-201: Another instance of possibly needing a range of values rather than just a Na number.

Edited to (Line 200-201): “...pickup ion gyroradius of $\sim 0.1 R_M$ (assuming the ions velocity is perpendicular to **B and is either Na^+ , Mg^+ , Al^+ , or Si^+).”**

Lines 204-210: This paragraph is still holding onto the idea that Na is behind all this, when it could be other species. The required neutral densities would be different for each of the species these pickup ions could be, because of the variations in ionization rate, so citing the density required at a given altitude is different for each species. Greater credence to the assumptions of these being Na^+ and Si^+ would be gained by demonstrating that neither Mg nor Al results in believable densities at that altitude. Regardless of how this is handled, this paragraph needs to be made more general for the group of possible ions. This is touched upon in the Methods section, though more needs to be done there to describe the results (see additional comments below).

We have made this paragraph less sodium focused. We have still included the discussion of the sodium exosphere, as a way for the reader to be able to compare the values. The paragraph now reads (Lines 206-214):

“To observe these fluxes we estimate that the required neutral density is expected to be $\sim 10^2 \text{ cm}^{-3}$ **if we assume the ions detected are Na^+ , $10^1 - 10^2 \text{ cm}^{-3}$ if we assume Si^+ , and $\sim 10^0 \text{ cm}^{-3}$ if we assume Al^+** (see **Methods** for more details). **For all species these are values that would not be expected to be observed at such large altitudes.** For sodium, these values are similar to sodium exospheric densities found at $\sim 700 \text{ km}$ altitude (for the high density subsolar point, Cassidy et al., 2015; 2016), and therefore at $\sim 5300 \text{ km}$ altitude the “global” exosphere cannot account for such high sodium densities **(if we assume a composition solely of Na^+ in our measurements)**. To explain this observation, requires a non-typical interpretation.”

I realize that the above bits of text were generated because of the original focus on Na, but it would be little effort to rework these few sections to be more representative of the true situation without having to redo any significant reanalysis.

We agree with the reviewer and have edited these areas as suggested. We have also updated the “Estimating Neutral Densities” section, Lines 405 – 435 in Methods) to reflect this and to keep the paper consistent.

Lines 266-273: The arguments for Na⁺ and Si⁺ dominating are based on the assumption of *atoms* being released from the surface. There is a growing body of evidence that impacts can release molecules that subsequently photodissociate to the component atoms. It is beyond the scope of this paper to delve into this issue; however, it is worth a sentence or two to note that it happens. I suggest something like this right after the point about the Al photoionization rate:

“However, we note that there is growing evidence for the possibility that micrometeoroid impacts release molecules that subsequently photodissociate into the neutral atoms that are observed at higher altitudes (e.g., for the Ca exosphere, Killen and Hahn, 2015). Should such a process apply to Al, this species could also comprise some fraction of the observed heavy ions.”

We thank the reviewer for this suggestion. After careful consideration, we think that although this process is possible, we do not think that it is viable for Al because of two reasons. 1) the AIO will be more massive, and therefore it will be slower, and therefore it will dissociate before reaching high altitudes (the photodissociation lifetime is 2000s from Berezhnoy & Klumov 2008) ; 2) coupled with a very low photoionization lifetime this Al atom will quickly ionize after dissociation, before reaching high altitudes. For Si, this process maybe more viable, since the photodissociation lifetime is much lower (600s). However, as the reviewer states, venturing into this topic in depth is beyond the scope of the paper. We have therefore discussed it very briefly in the Methods section, and our conclusion is that the most likely composition is Si and Na.

We adapted the short text that the reviewer wrote and now the current paragraph in the methods section reads (Line 466-484):

“We also note that there is evidence for the possibility that micrometeoroid impacts release molecules that subsequently photodissociate into the neutral atoms that are observed at higher altitudes (e.g., for the Ca exosphere, Killen et al., 2005; Killen and Hahn, 2015). This may be an important process in regards to both Al and Si, with the photodissociation of AIO and SiO (e.g. Berezhnoy, 2018), and therefore we briefly discuss it here. Berezhnoy & Klumov (2008) investigated the photodissociation lifetimes of these molecules and found them to be 2000 and 600 s (AIO and SiO, respectively). Due to the increased mass of the particle, these particles will have a lower velocity and will be photodissociated before reaching higher altitudes, in comparison to the atoms. After photodissociation, Al still has a very short photoionization lifetime, and so it will be ionized quickly after photodissociation. Therefore, we do not think this is a viable method for Al to reach high altitudes to be observed by FIPS – it will not reach extremely high altitudes in 2000 s as a molecule, and then it will be quickly ionized as an atom and trapped in the magnetosphere. This however, may be a more viable method for Si, because of the lower photodissociation lifetime (the molecule will photodissociate shortly after ejection from the surface), and with the longer photoionization lifetime it may be able to reach higher altitudes as an atom. However, investigating this process in depth is beyond the scope of this paper, and we do not consider it further.”

I note that in lines 452-459, this argument is already mentioned in a different context for Si. If the argument can be applied to limit Si, it can also be applied to enhance Al, so the text here and in the supplemental section should be changed to qualitatively reflect the possibility of an Al contribution. I agree that Mg is less likely to be a constituent as the dominant factor for Mg is the long photoionization time.

This has now been changed due to the above comment.

General Comments

I suggest replacing all references in the main text to a figure “panel x” with “Figure Nx” so that it is clear which figure is being referenced. Although some instances are close enough to the original figure reference to be reasonably clear, not all occurrences throughout the paper are.

Done.

In the figure captions, the use of a single) with the panels is awkward in appearance. I suggest using two, for example (a) instead of a), in all figure captions.

Done.

Specific Comments

Line 18: Suggest changing “which contains” to “with measured”.

Done. We agree that this is a lot better.

Line 25: Need a space between 700 and km.

Done.

Line 40: Need a space between 450 and km.

Done.

Line 48: Replace the “st” after 21 with a comma.

Done.

Figure 2 caption: Replace the “st” after 21 with a comma.

Done.

Figure 3 caption: References to “sodium ions” need to be changed to “sodium-group ions”. There are three of them.

We have changed the first one to “Sodium-group pickup ions” as suggested and changed the other two to “pickup”.

Line 145: Add “26P/” in front of “Grigg”. Although there may not be another comet Grigg-Skjellerup, this is the standard name of this comet.

Done.

Line 152: Change “instruments” to “instrument’s”.

Done.

Figure 4 caption: Change “extended sodium exosphere event” to “transient exosphere event”.

Done.

Figure 4: The idea that these heavy ions are mostly Na⁺ and Si⁺ has not been introduced yet, so I suggest that the “Na” and “Si” in the figure be replaced with “Neutral atom” and the “Na⁺” and “Si⁺” with “Ion”, and the caption changed accordingly. This will work for whatever the composition of the heavy ions ends up being.

This suggestion has been implemented, and this (formerly Figure 4c) is now Figure 5.

Line 206: Change “methods” to “Methods”.

Done.

Line 213: Delete “sodium” to keep this general to all species.

Done (now line 217).

Line 218: This number is for Na (presumably), but I suspect it is close enough to say that it applies to all the sodium-group ions. The point is that thermal desorption is not a player here, and that will hold for all of the possible species.

It is for Na. Considering that the previous mention of Na is deleted in this paragraph, and the ~1.5 eV isn’t specified as sodium, (that we agree with the reviewer that this close enough to hold for all the species), we have left this sentence as is.

Line 220: Change “the sodium” to “an atom” to keep this general.

Done.

Lines 225-227: It is fine to keep this as written, because Na is just given as an example.

Kept as suggested.

Line 251: Before the sentence beginning “The impact...”, I suggest adding the following.

“In contrast, impact vaporization has been postulated as the dominant source of both the Ca and Mg exospheres at Mercury [Burger et al., 2014; Merkel et al., 2016].”

Sentence added as suggested, and the references. We assume the reviewer meant Merkel et al., (2017) – it has a doi as 2016 but was published in Jan 2017:

Merkel, A. W., Cassidy, T. A., Vervack, R. J., McClintock, W. E., Sarantos, M., Burger, M. H., & Killen, R. M. (2017). Seasonal variations of Mercury's magnesium dayside exosphere from MESSENGER observations. *Icarus*, 281, 46– 54. <https://doi.org/10.1016/j.icarus.2016.08.032>

Lines 251-253: Change the beginning of the sentence “The impact...” to be “In any case, an impact will vaporize the sodium-group species with high enough...”. This will keep the argument general.

Done.

Line 258: Need a space between 1.5 and m.

Done.

Line 260: Change “exospheric” to “sodium exospheric”.

Done.

Line 260: Change “1m meteorite” to “1-m meteoroid”.

Done.

Line 265: Change “neutral densities” to “neutral sodium densities”.

Done.

Lines 277-278: The reference to Figure 4c (the first for that panel, I think) is far enough away from the Figure 4 discussion and is more qualitative than the rest of Figure 4 that this could conceivably be a Figure 5 and located nearer to this part of the text.

As suggested, we have separated Figure 4 into two parts. There is now a Figure 5.

Line 318: Need to add Al⁺ to the list of Na⁺-group ions.

Done.

Line 367: Change “sodium” to “sodium-group”.

Done.

Line 392: Change “sodium” to “sodium-group”.

Done.

Lines 398-401: Suggest moving these lines to the end of the section (lines 421-425) and keeping this section general to any species. At the end of the section, these can be combined with lines 421-425 to provide an example for sodium.

We have edited it differently, but made the section “Estimating Neutral Densities” less sodium focused.

Reviewer #2 (Remarks to the Author):

The authors answer properly to my comments and changed the paper accordingly. I therefore recommend this paper for publication. I have only few very minor comments which do not need a new review:

Line 37: I would replace “most abundant species” by “most abundant observed species”. There is a strong bias associated with the detection of Na atoms when using visible spectrometer. Therefore, I would remain careful regarding the real composition of Mercury’s exosphere.
Done.

Line 44: I do not understand the logic of this "Therefore". There is no relation between the sodium radiation pressure and Mercury's magnetosphere and therefore no reason to associate the two as it seems to be done.

**This has been explained a bit further, in regards to similar structures. Line 43-45:
“Similarly, the magnetosphere is compressed on the dayside by the solar wind, and the nightside magnetic field is stretched out to form a magnetotail, and so the exosphere mostly lies within the magnetosphere of Mercury.”**

Line 79-80: is this sentence related to sodium species needed here? I would suppress it to avoid any confusion regarding the sodium-group composition.

Line 83: same remark: I would replace “sodium” by “sodium-group”

These sentences are related to distinguishing sodium of planetary origin rather than solar wind origin. We have grouped these sentences now, and added the following “(which lies within this m/e range)” and mentioned the sodium-group earlier to underline that sodium is in this mass-to charge group and we are mentioning sodium here as an example.

Line 209: same remark than for Line 83.

We have added more text above this to make sure we discuss the other species and we also added the following caveat to this sentence to show that we are discussing sodium as an example: “(if we assume a composition solely of Na⁺ in our measurements).” (on Line 213)

Line 213: I would suppress “sodium”, since the discussion regarding the exosphere should be extended to other species. As a matter of fact, this whole paragraph (up to line 273) is really focused on the origin of the sodium in the exosphere. Since, the main conclusion is that the detection might be also due to other species, would it be not more relevant to change slightly this part by providing some information regarding the origins of the other exospheric species? As an example, lines 266-267 might surprise the reader since Al, Si and Mg are not discussed at all previously.

Reviewer #1 has made a similar comment. This section has now been edited in regards to suggestions from Reviewer #1 and is now more general (Line 216-282).

Line 455: Killen et al. (2005) explained the origin of the very large height scale of the exospheric Ca by suggesting that Ca was ejected as CaO and then dissociated by photo-dissociation providing energy to the dissociated component. Therefore, if Si is preferentially ejected as SiO a similar process could happen and actually favour the capacity of Si to reach 5300 km altitude.

We agree about this and have included text in the Methods section, in response to both your comment and Reviewer #1 similar comment. The text on (Line 468-486) reads:

“We also note that there is evidence for the possibility that micrometeoroid impacts release molecules that subsequently photodissociate into the neutral atoms that are observed at higher altitudes (e.g., for the Ca exosphere, Killen et al., 2005; Killen and Hahn, 2015). This may be an important process in regards to both Al and Si, with the photodissociation of AlO and SiO (e.g. Berezhnoy, 2018), and therefore we briefly discuss it here. Berezhnoy & Klumov (2008) investigated the photodissociation lifetimes of these molecules and found them to be 2000 and 600 s (AlO and SiO, respectively). Due to the increased mass of the particle, these particles will have a lower velocity and will be photodissociated before reaching higher altitudes, in comparison to the atoms. After photodissociation, Al still has a very short photoionization lifetime, and so it will be ionized quickly after photodissociation. Therefore, we do not think this is a viable method for Al to reach high altitudes to be observed by FIPS – it will not reach extremely high altitudes in 2000 s as a molecule, and then it will be quickly ionized as an atom and trapped in the magnetosphere. This however, may be a more viable method for Si, because of the lower photodissociation lifetime (the molecule will photodissociate shortly after ejection from the surface), and with the longer photoionization lifetime it may be able to reach higher altitudes as an atom. However, investigating this process in depth is beyond the scope of this paper, and we do not consider it further.”

Reviewer #3 (Remarks to the Author):

The manuscript has greatly improved over the previous version. I still have problem with the title, it is not an exospheric observation. It is an ion measurement that infers the exosphere, so I would substitute the word "observed" with "inferred", as in the abstract and as suggested by reviewer 1.

As I wrote in the previous review, the observation, unique in the MESSENGER data, is surely of great interest and deserves publication. The interpretation of such event will surely open new views of the Mercury's environment. Finally, I recommend the publication after the title change.

Thank you for your comments. We have changed the title to use the word "inferred" rather than "observed" as the reviewer has suggested.

REVIEWERS' COMMENTS:

Reviewer #1 (Remarks to the Author):

The authors have done a good job of revising the paper to be what this reviewer considers a more likely interpretation of the data. I recommend this for publication, and I appreciate the patience they have shown in waiting for reviews during these uncertain times.

Minor comments:

Line 135: There is a missing symbol or something after "injection site".

Lines 152-154: The word "energy" is used a lot. Maybe reword sentence to be less repetitious.

Line 199: "ions" should be "ion's"; same sentence, insert "the ion" between "and" and "is".

Line 209: "high density" should be "high-density".

Line 261: "the crossing" should be "a crossing" as Mercury crosses the Encke dust trail three times depending on the TAA and the particle size.

Reviewer #2 (Remarks to the Author):

The revised version answered properly to my comments and is ready for publication.

We thank the Reviewers for their review of our paper. We have responded in bold to the referees comments below.

**Best Regards,
Jamie M. Jasinski (on behalf of all the authors).**

REVIEWERS' COMMENTS:

Reviewer #1 (Remarks to the Author):

The authors have done a good job of revising the paper to be what this reviewer considers a more likely interpretation of the data. I recommend this for publication, and I appreciate the patience they have shown in waiting for reviews during these uncertain times.

Minor comments:

Line 135: There is a missing symbol or something after “injection site”.

We have changed this to “star”.

Lines 152-154: The word “energy” is used a lot. Maybe reword sentence to be less repetitious.

We have deleted the 2nd energy.

Line 199: “ions” should be “ion’s”; same sentence, insert “the ion” between “and” and “is”.

Done

Line 209: “high density” should be “high-density”.

Done.

Line 261: “the crossing” should be “a crossing” as Mercury crosses the Encke dust trail three times depending on the TAA and the particle size.

Done

Reviewer #2 (Remarks to the Author):

The revised version answered properly to my comments and is ready for publication.

Thank you.